# XPO5 promotes primary miRNA processing independently of RanGTP

Jingjing Wang [1], Jerome E. Lee[1,3], Kent Riemondy[1,4], Yang Yu[2], Steven M. Marquez[1], Eric C. Lai [2] & Rui Yi [1✉]

XPO5 mediates nuclear export of miRNA precursors in a RanGTP-dependent manner. However, XPO5-associated RNA species have not been determined globally and it is unclear whether XPO5 has any additional functions other than nuclear export. Here we show XPO5 pervasively binds to double-stranded RNA regions found in some clustered primary miRNA precursors and many cellular RNAs. Surprisingly, the binding of XPO5 to pri-miRNAs such as *mir-17~92* and *mir-15b~16-2* and highly structured RNAs such as vault RNAs is RanGTP-independent. Importantly, XPO5 enhances the processing efficiency of *pri-mir-19a* and *mir-15b~16-2* by the DROSHA/DGCR8 microprocessor. Genetic deletion of *XPO5* compromises the biogenesis of most miRNAs and leads to severe defects during mouse embryonic development and skin morphogenesis. This study reveals an unexpected function of XPO5 for recognizing and facilitating the nuclear cleavage of clustered pri-miRNAs, identifies numerous cellular RNAs bound by XPO5, and demonstrates physiological functions of *XPO5* in mouse development.

[1] Department of Molecular, Cellular, and Developmental Biology, University of Colorado Boulder, Boulder, CO, USA. [2] Department of Developmental Biology, Memorial Sloan Kettering Cancer Center, 1275 York Avenue, Box 252, New York, NY, USA. [3]Present address: ArcherDX, Boulder, CO, USA. [4]Present address: RNA Bioscience Initiative, University of Colorado School of Medicine, Aurora, CO, USA. ✉email: yir@colorado.edu

XPO5 is a karyopherin protein[1] that mediates nuclear export of miRNA hairpin precursors (pre-miRNAs)[2–4]. XPO5 binds to hairpin RNAs with 2–8 nt 3′ overhang including pre-miRNA and minihelix viral RNA in a RanGTP-dependent manner[2–5]. The formation of XPO5:pre-miRNA:RanGTP nuclear export complex was confirmed by X-ray crystallography[6], in which RanGTP binding triggers a conformational change of XPO5 and several XPO5 residues in the HEAT repeats form hydrogen bonds with 5′ and 3′ ends including the 2-nt 3′ overhang of pre-miRNAs. Functional studies showed a requirement of XPO5 to transport pre-miRNAs from the nucleus to the cytoplasm[2–4], providing the evidence that XPO5 is an essential component of miRNA biogenesis[7]. However, XPO5-independent pre-miRNA export pathway was also reported[8,9]. Notably, 7-methylguanosine-capped pre-miRNAs whose biogenesis is independent of the DROSHA/DGCR8 microprocessor are exported via the PHAX-XPO1 pathway[9]. In a recent study, deletion of XPO5 in human HCT116 cells only mildly affected the biogenesis of most miRNAs, suggesting that XPO5 is sufficient but not required for miRNA biogenesis[10]. However, in a CRISPR-Cas9 screen for genes that are critical for mir-19-mediated silencing also in human HCT116 cells, XPO5 was identified as essential[11]. In addition to these different reports, the lack of genetically engineered mouse models of XPO5, unlike other core components of the miRNA pathway such as Drosha, Dgcr8, Dicer1, or Ago genes, has hindered the understanding of XPO5 functions. Overall, the in vivo requirements of XPO5 for global miRNA biogenesis, mammalian development, and tissue formation remain to be determined.

XPO5 is a highly expressed protein among core components of miRNA biogenesis in both mouse and human cells[12]. However, it is more sensitive to saturation caused by the expression of short hairpin RNA and highly structured viral RNA than other components such as Drosha and Dicer1[13,14]. Although individual RNA species such as pre-miRNA[2–4], some cellular tRNAs[15] and minihelix viral RNAs such as adenovirus VA1 RNA[5,14] have been identified as XPO5 cargoes for nuclear export, XPO5-associated cellular RNAs have not been determined at the genomic scale. It is unclear how many pre-miRNAs directly interact with XPO5 and require XPO5 for their biogenesis. Furthermore, since the discovery of XPO5 as the nuclear export factor for pre-miRNA hairpin[2–4,6], much attention to XPO5 has been drawn to its nuclear export functions. However, XPO5 was originally identified as a binding protein of double-stranded RNA (dsRNA) binding proteins such as ILF3, PKR, and Staufen[16]. Although the direct binding of RNA species including pre-miRNAs, tRNA and VA1 RNA to XPO5 has since been well documented[2–6,14,15], it remains unknown whether other cellular RNAs with double-stranded regions can bind to XPO5. In addition, all known RNA substrates of XPO5 bind to XPO5 in a RanGTP-dependent manner, which is a key feature of karyopherin-mediated nuclear export[1]. As a result, whether XPO5 plays any roles in cellular RNA metabolism other than nuclear export is an open question.

To address these questions, we first perform HITS-CLIP analysis of XPO5-associated RNAs in human embryonic kidney cells 293T (HEK293T). We find that a large number of cellular RNAs including most pre-miRNAs and numerous noncoding, structural RNAs are bound by XPO5. Notably, some closely clustered primary miRNA precursors such as pri-mir-17~92 and pri-mir15b~16-2 show strong XPO5 HITS-CLIP signals outside of pre-miRNA hairpins. Surprisingly, purified XPO5 directly bind to pri-mir-17~92 and pri-mir15b~16-2 in a RanGTP-independent manner. In vitro processing assays reveal that XPO5 enhances the cleavage efficiency of pri-mir-19a and pri-mir-15b~16-2 by the DROSHA/DGCR8 microprocessor.

To test the function of XPO5 in mouse development, we show that constitutive deletion of XPO5 leads to early embryonic lethality and failed gastrulation at approximately embryonic day 7.5. For tissue morphogenesis, we find that conditional knockout (cKO) of XPO5 in the epithelial cells of the skin shows similar but slightly different defects than those observed in Dicer1 or Dgcr8 cKO[17–19]. Quantitative measurement of miRNA biogenesis in the skin shows ~90% global loss for most canonical miRNAs except for a few Drosha/Dgcr8-independent miRNAs such as mir-320 and mir-484[19]. Taken together, this work provides insights into the function of XPO5 in miRNA biogenesis that is independent of RanGTP binding and nuclear export, and identifies numerous cellular RNAs as XPO5 binding partners. These findings establish a molecular basis to further investigate the regulatory roles of XPO5 in miRNA biogenesis and the metabolism of other cellular RNAs.

## Results

**XPO5 associates with primary and pre-miRNA precursors**. UV crosslinking followed by immunoprecipitation and sequencing has been successfully used to identify highly structured miRNA precursors that are bound by DROSHA, DGCR8, and DICER1[20–22]. To globally identify XPO5-associated RNA species in human cells, we applied a high-throughput sequencing of RNA isolated by crosslinking immunoprecipitation (HITS-CLIP) approach to map RNA fragments bound by endogenous XPO5 in HEK293T cells (Fig. 1a). In the absence of XPO5 antibody, we did not recover any RNA fragments upon immunoprecipitation (Fig. 1a). Five independent HITS-CLIP libraries were generated and they displayed generally consistent profiles of XPO5-associated RNAs. To increase stringency, we required the detection of any RNA fragments in at least three out of the five libraries for further analysis. In total, 64.2 million reads were generated and mapped to 83,169 genomic peaks (see "Methods"). Because XPO5 is known to interact with pre-miRNA hairpin and mediates their nuclear export[2–4], we first determined whether pre-miRNA hairpins were recovered in our datasets. Indeed, all 37 pre-miRNAs of the most highly expressed miRNAs, judging from miRNA reads number (>1000 reads) from a quantitative smRNA-seq in the HEK293T cells (Supplementary Data 1), were detected in XPO5 HITS-CLIP (Fig. 1b). Overall, 96 out of the top 107 miRNAs, whose read number was >200 in the smRNA-seq, were also detected in our XPO5 HITS-CLIP. Furthermore, XPO5-associated miRNA sequences generally spanned the left arm, the top loop, and the right arm regions, reflecting the association of XPO5 with pre-miRNA hairpins prior to DICER1 cleavage. Only ~20% of XPO5 HITS-CLIP reads were mapped to either arm (Fig. 1c), which generally represents mature miRNAs. In comparison, >98% of mature miRNA reads recovered from smRNA-seq were mapped to the left or the right arm as expected (Fig. 1c).

To compare the profile of XPO5-associated miRNA sequences with other essential components of the miRNA biogenesis pathway including DROSHA, DGCR8, DICER1, and AGO proteins, we downloaded previous published HITS-CLIP (DROSHA and DGCR8)[20,21], PAR-CLIP (DICER1) and AGO2/3-Seq (mature miRNAs)[22] datasets and examined the profile of miRNA sequences that are associated with each component (Fig. 1d–h). To facilitate the comparison of all miRNA profiles captured by each CLIP dataset, we aligned all detected miRNA sequences starting from 5′ end of 5p miRNAs as annotated in miRBase release 21[23]. Overall, DROSHA and DGCR8 HITS-CLIP showed similar profiles, which usually span not only the pre-miRNA hairpin regions but also the flanking regions. In agreement with the notion that DROSHA recognizes the lower stem from the basal junction that is outside of the pre-miRNA

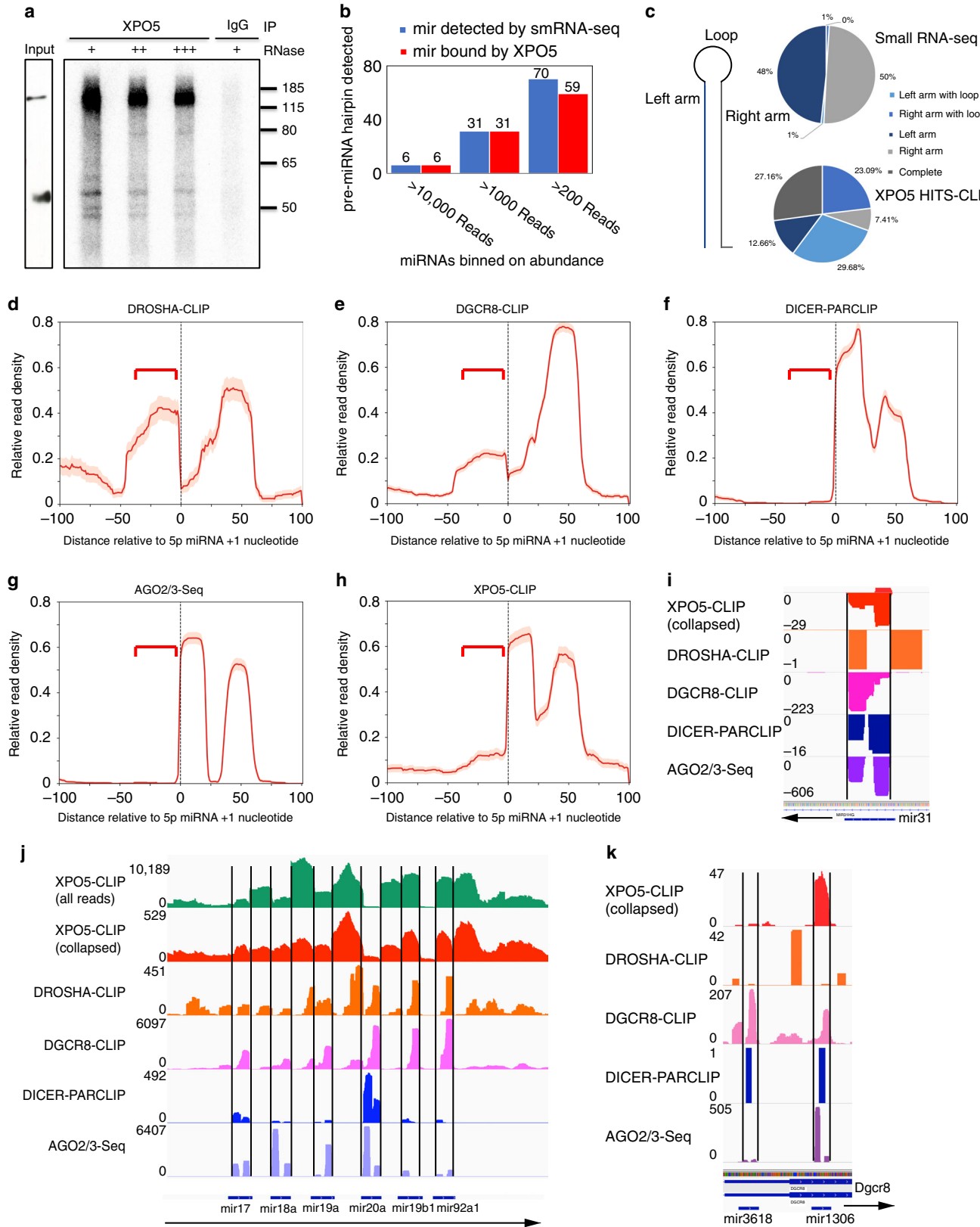

hairpin and DGCR8 recognizes the upper stem of the miRNA hairpin[24], DROSHA-associated miRNA sequences showed more prominent signals in the sequences immediately upstream of pre-miRNA hairpin than DGCR8-associated miRNA sequences (Fig. 1d, e). DICER1 PAR-CLIP showed the recognition of pre-miRNA hairpin but not the lower stem regions outside of

pre-miRNA hairpin (Fig. 1f), consistent with that DICER1 recognizes pre-miRNA hairpin after its release by DROSHA/DGCR8. AGO2/3-Seq showed the profile of mature miRNAs (Fig. 1g), which are derived from either the left or the right arm of miRNA hairpin. However, XPO5 HITS-CLIP showed not only the association of XPO5 with pre-miRNA hairpin as expected but

**Fig. 1 HITS-CLIP reveals association of XPO5 with primary and pre-miRNA precursors. a** Autoradiogram of [32]P-labelled XPO5-RNA complexes treated with different concentration of RNase is resolved on a PAGE gel. Original data are provided in the Source Data file. **b** Pre-miRNAs of most abundantly expressed miRNAs in HEK293T cells are detected in XPO5 HITS-CLIP. **c** The location of small RNA-seq reads and XPO5 HITS-CLIP reads relative to miRNA hairpin is shown in pie charts, respectively. Light blue indicates reads covering both the left arm and the loop region; blue indicates reads covering both the right arm and the loop region; dark blue indicates reads covering only the left arm; light gray indicates reads covering only the right arm; dark gray indicates reads covering the entire hairpin. **d–h** Metagene analysis of miRNA sequences associated by DROSHA, DGCR8, DICER1, AGO2/3, and XPO5 reveals the recognized regions by each core component and identifies pri-miRNA sequences associated by DROSHA, DGCR8, and XPO5 that are beyond pre-miRNA hairpin (red bracket). **i, k** IGV tracks of XPO5-CLIP, DROSHA-CLIP, DGCR8-CLIP, DICER1-PARCLIP, and AGO2/3-Seq reveal the specific association of XPO5 with *pre-mir-31*, *pre-mir-3618*, and *pre-mir-1306*. Negative reads number in IGV track indicates the mapping to the minus strand. **j** IGV tracks of XPO5-CLIP, DROSHA-CLIP, DGCR8-CLIP, DICER1-PARCLIP, and AGO2/3-Seq reveal the widespread association of XPO5 with *pri-mir-17~92*. Blue bars at the bottom and black lines together indicate the location of pre-miRNAs. Data range is shown on the left of each track.

also the flanking regions of the hairpin (Fig. 1h). Comparing the overall profile of XPO5-associated miRNA sequences to those of DICER1 and DGCR8, it was evident that XPO5-associated miRNA regions were similar to DICER1 within the annotated pre-miRNA hairpins but also resembled the profile of DGCR8-associated regions outside of the annotated pre-miRNAs (Fig. 1e, f, h).

To understand how XPO5 associates with miRNA precursors outside of pre-miRNA hairpin, we next examined XPO5 HITS-CLIP data of individual miRNAs. To faithfully present the data, we used one IGV track to show all reads mapped from XPO5 HITS-CLIP data and another track to show all unique reads after collapsing potential duplicates (Fig. 1i–k and Supplementary Fig. 1a, b). While many miRNAs are encoded by a primary transcript individually, ~50% of miRNAs are also encoded by polycistronic transcription unit[25,26]. For all singly encoded miRNAs that we have examined such as *mir-31, mir-21*, and *mir-34a*, XPO5 HITS-CLIP signals were faithfully restricted within pre-miRNA hairpins as expected (Fig. 1i and Supplementary Fig. 1a, b). However, for many polycistronic miRNAs especially the ones that are closely clustered, XPO5 HITS-CLIP reads also covered primary miRNA transcripts. Among all XPO5-associated miRNAs, the *mir-17~92* cluster harbored the most reads. Remarkably, XPO5 HITS-CLIP reads broadly covered the pri-miRNA, distinct from the profile of pre-miRNA precursors as detected in DICER1 PAR-CLIP (Fig. 1j). Inter-pre-miRNA regions between *mir-17* and *mir-19b1* all had abundant XPO5-associated RNA reads compared with those derived from pre-miRNA hairpins (Fig. 1j). Upon closer inspection, we noted two possible modes of XPO5 binding to *pri-mir-17~92*, judging by the distinct reads density covering the precursor regions. First, XPO5 bound to pre-miRNA hairpins of each of these six miRNAs. Second, XPO5 bound to inter-pre-miRNA regions with an even higher density than the pre-miRNA regions. For example, both mir-19a and mir-20a are abundantly expressed miRNAs. But XPO5-associated RNA reads were far more abundant in the inter-pre-miRNA regions than the pre-miRNA regions. Furthermore, many XPO5-associated RNA reads span the predicted DROSHA cleavage sites that mark the 5′ and 3′ ends of pre-miRNAs (Supplementary Fig. 1c), indicating the association of XPO5 with uncleaved pri-miRNAs. The only inter-pre-miRNA region that is depleted of XPO5-associated RNA reads was between *mir-19b1* and *mir-92a1*. Notably, *mir-17~19b1* pri-miRNA is released from the *mir-17~92* pri-miRNA by an endonuclease CPSF3 and the spliceosome-associated ISY1[27]. These data suggest a possibility that XPO5 recognizes *pri-mir-17~92* cluster after the CPSF3/ISY1-mediated cleavage but prior to DROSHA/DGCR8-mediated processing.

To confirm that the association of XPO5 to pri-miRNAs is specific and not simply due to the abundance of *pri-mir17~92*, we inspected other closely clustered pri-miRNAs such as both *mir-15a~16-1* and *mir-15b~16-2* clusters that are expressed at a much

lower level than *pri-mir-17~92*. We still observed prominent inter-pre-miRNA reads in XPO5 HITS-CLIP data and the distinct reads density in the inter-pre-miRNA and pre-miRNA regions (Supplementary Fig. 1d). Interestingly, the widespread XPO5 reads coverage was not found in other miRNA clusters, in which pre-miRNA hairpins are located far away from each other such as the *mir-200b* cluster (>600 nt between hairpins) (Supplementary Fig. 1e). These data suggest that not all primary transcripts of polycistronic miRNAs bind to XPO5. In addition, two miRNAs, *mir-3618* and *mir-1306*, are processed from the 5′ region of *Dgcr8* mRNA[28]. However, the expression of *mir-1306* is much higher than *mir-3618* based on both the processing assay[28] and the sequencing of AGO2/3 associated mature miRNAs[22]. Consistent with the pattern of mature miRNA expression, XPO5 HITS-CLIP reads on *mir-1306* pre-miRNA were much more abundant than those on *mir-3618* pre-miRNA hairpin (Fig. 1k), lending further support to the specificity of our XPO5 HITS-CLIP. Together, these data suggest two modes of XPO5 association with miRNA precursors: (1) for most miRNAs, XPO5 associates with pre-miRNA hairpins as expected; and (2) for some closely clustered polycistronic miRNAs, XPO5 associates with the primary miRNA transcripts with a high affinity in inter-pre-miRNA regions.

**XPO5 binds to pri-miRNA precursors independently of RanGTP.** To study the binding of XPO5 to closely clustered primary miRNAs, we performed electrophoretic mobility shift assays in vitro. We purified human XPO5 and RanQ69L, a Ran mutant that is deficient in GTP hydrolysis and locks Ran in the RanGTP form[29], from bacteria *E.coli* as described previously[4]. We first used in vitro transcribed *pre-mir-30a* hairpin RNA to confirm the binding activity of recombinant XPO5 and RanQ69L proteins to pre-miRNA. Indeed, XPO5 binds to *pre-mir-30a* hairpin in a RanGTP-dependent manner (Fig. 2a). The single-shifted band also confirmed the formation of XPO5:RanGTP:*pre-mir-30a* complex. Because *pri-mir-17~92* is known to fold into a highly structured RNA conformation that has extensive base-pairing regions beyond pre-miRNA hairpins[27] (Supplementary Fig. 2a), we in vitro transcribed and folded the *pri-mir-17~92* precursor (Supplementary Fig. 2b) and tested its binding to XPO5. Surprisingly, XPO5 bound to *pri-mir-17~92* in a RanGTP-independent manner (Fig. 2b). Inclusion of RanQ69L together with XPO5 and *pri-mir-17~92* showed no interference to the binding of XPO5 and the pri-miRNA (Fig. 2c). Neither the size nor the intensity of the super-shifted bands containing XPO5 and *pri-mir-17~92* changed in the presence or absence of RanQ69L (Fig. 2c). This result suggests that the complex does not contain RanQ69L. In addition, we observed continuously super-shifted bands with the increasing doses of XPO5, up to 80-fold of XPO5 in excess of *pri-mir-17~92* RNA (Fig. 2b, c). These data suggest that multiple XPO5 molecules bind to *pri-mir-17~92* precursor in a RanGTP-independent manner in vitro, corroborating with the

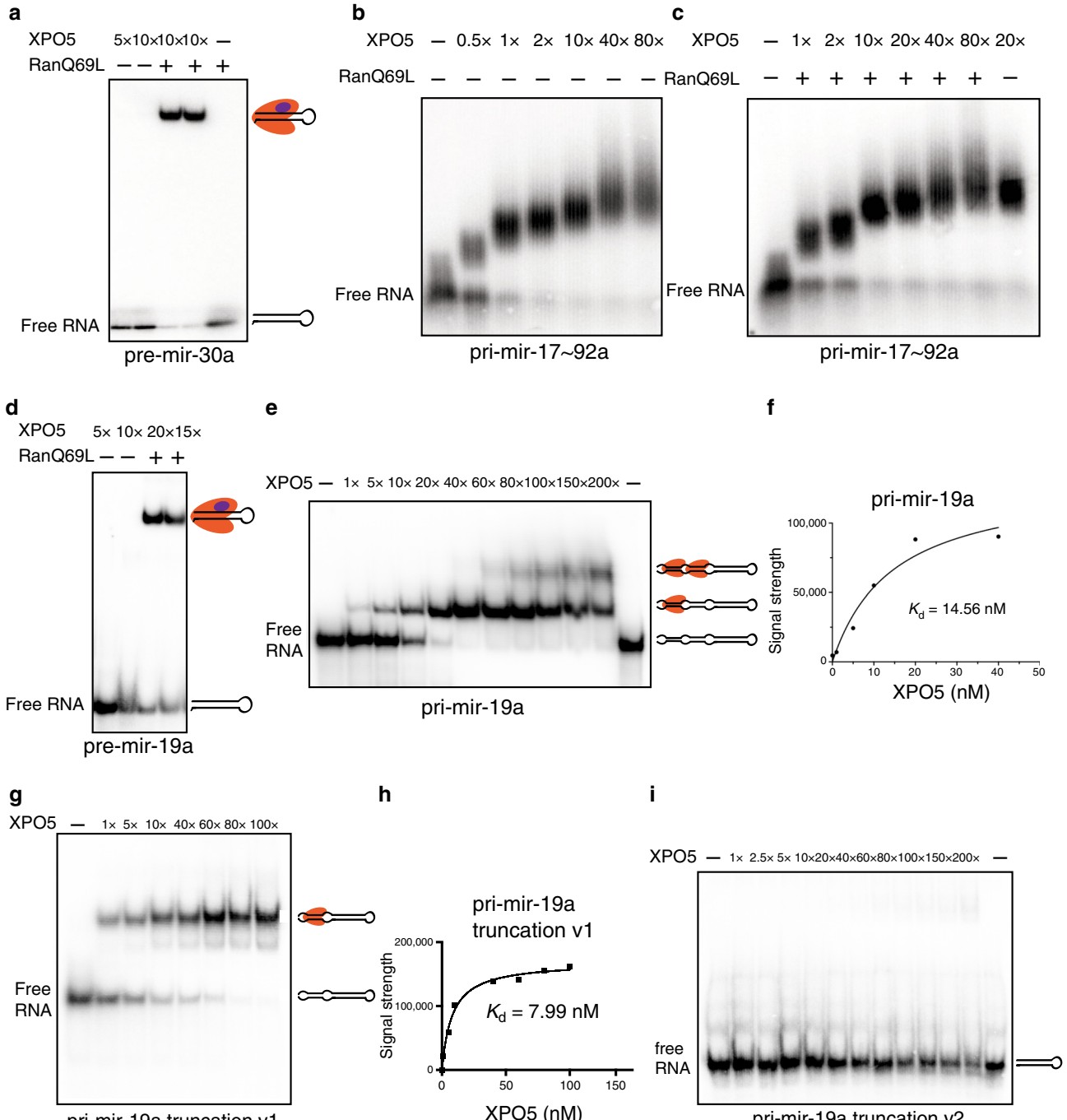

**Fig. 2 XPO5 binds to primary miRNA precursors in a RanGTP-independent manner. a** XPO5 binds to *pre-mir-30a* in a RanGTP-dependent manner. Black hairpin represents *pre-mir-30a*, XPO5 is coloured in orange, RanGTP is coloured in purple. **b** XPO5 binds to *pri-mir-17~92* without RanGTP. Increasing amount of XPO5 results in super shifts. **c** XPO5 binding to *pri-mir-17~92* is not affected by RanGTP. **d** XPO5 binds to *pre-mir-19a* in a RanGTP-dependent manner. Black hairpin represents *pre-mir-19a*, XPO5 is coloured in orange, RanGTP is coloured in purple. **e** XPO5 binds to *pri-mir-19a* in a RanGTP-independent manner. Increasing amount of XPO5 results in a super shift. Black hairpin represents *pri-mir-19a*, XPO5 is coloured in orange. **f** The dissociation constant (Kd) between XPO5 and *pri-miR-19a* is calculated based on the binding results in **e**. **g** XPO5 binds to truncated *pri-mir-19a* v1 (*pri-mir-19a* truncation v1) in a RanGTP-independent manner. Black hairpin represents *pri-mir-19a* truncation v1, XPO5 is coloured in orange. **h** The dissociation constant (Kd) between XPO5 and *pri-miR-19a* truncation v1 is calculated based on the binding results in **g**. **i** XPO5 does not bind to truncated *pri-mir-19a* v2 (*pri-mir-19a* truncation v2). For all binding assays, 1 nM substrates are used. Original data for **a–e**, **g**, and **i** are provided in the Source Data file.

extensive XPO5 HITS-CLIP signals detected on *pri-mir-17~92* in vivo.

To further characterize how XPO5 binds to primary miRNA transcript, we focused on *pri-mir-19a*, whose flanking sequences are predicted to form extensive base-pairing outside of the 11 bp lower stem region that is characteristic of DROSHA/DGCR8 primary miRNA substrates[7] (Supplementary Fig. 2c). The primary miRNA sequences flanking *mir-19a* also had very high XPO5 HITS-CLIP signals (Fig. 1j). We first confirmed that *pre-mir-19a* hairpin only bound to XPO5 in a RanGTP-dependent

manner (Fig. 2d). In contrast, *pri-mir-19a* bound to XPO5 in a RanGTP-independent manner, similar to the *pri-mir-17~92* results. Importantly, two distinct *pri-mir-19a*:XPO5 complexes, judging by the size of super-shifted *pri-mir-19a*, were detected with the increasing amount of XPO5 (Fig. 2e). Finally, the dissociation constant (Kd) of *pri-mir-19a* and XPO5 association was calculated at 14.56 nM (Fig. 2f).

Because *pre-mir-19a* hairpin could not bind to XPO5 in the absence of RanGTP, these results suggested that XPO5 binds to the lower stem regions of *pri-mir-19a*. To test this hypothesis, we generated two truncated *pri-mir-19a* mutants. We deleted one basal stem region in *pri-mir-19a* truncation v1 and both basal stem regions in *pri-mir-19a* truncation v2 (Supplementary Fig. 2c). Indeed, *pri-mir-19a* truncation v1 bound to XPO5 in a RanGTP-independent manner with the Kd of 7.99 nM but only a singly shifted band was detected (Fig. 2g, h). In contrast, *pri-mir-19a* truncation v2 no longer bound to XPO5 (Fig. 2i) when both basal stem regions were deleted. These data suggest that the RanGTP-independent binding of *pri-mir-19a* by XPO5 requires the lower stem regions outside of the pre-miRNA hairpin.

**XPO5 facilitates the microprocessor cleavage of pri-miRNAs.** To determine the function of XPO5 binding to *pri-mir-19a* precursors, we performed the DROSHA/DGCR8-mediated processing assay for *pri-mir-19a* and the two mutants in vitro. Flag-tagged DROSHA and DGCR8 expression plasmids were co-transfected into HEK293T cells to obtain the microprocessor complex using immunoprecipitation and purification as described previoulsy[30]. We also purified Flag-XPO5 protein from transfected HEK293T cells. Western blot and silver stain indicated that DROSHA/DGCR8 microprocessor does not copurify with XPO5 (Supplementary Fig. 3a, b). In the absence of XPO5, the microprocessor complex released *pre-mir-19a* hairpin from the primary transcript with a modest efficiency (Fig. 3a, lane 3), in line with previously studies[30,31]. Notably, preincubation of *pri-mir-19a* with an increasing amount of XPO5 drastically enhanced the processing efficiency. The improved processing was demonstrated by both the significant reduction of unprocessed *pri-mir-19a* and approximately twofold more accumulation of *pre-mir-19a* in an XPO5 dosage-dependent manner (Fig. 3a, lanes 4–6). XPO5 alone did not alter *pri-mir-19a* (Fig. 3a, lane 7), ruling out the possibility of microprocessor contamination.

XPO5 has been well characterized for its binding to pre-miRNA hairpin that is dependent upon RanGTP and 2-nt 3′ overhang of pre-miRNA[6,32]. So far, we have showed the RanGTP-independent binding of pri-miRNA by XPO5 (Fig. 2b–e, g). We next asked whether XPO5 residues that mediate the interaction between XPO5 and the 5′ and 3′ ends of pre-miRNA are required for enhancing the microprocessor cleavage. To this end, we generated an XPO5 mutant with E445A, S606A, E711A, and R718A quadruple mutations. These residues were found to form hydrogen bonds between XPO5 HEAT repeats and the 5′ and 3′ ends of *pre-mir-30a* including the 2-nt 3′ overhang[6]. In support of the notion that XPO5 interaction with pri-miRNA is independent of its association with pre-miRNA, the XPO5 mutant retained the ability to potentiate the DROSHA/DGCR8 processing (Fig. 3a, lanes 8–11).

To further confirm that the ability of XPO5 to promote *pri-mir-19a* processing requires RanGTP-independent binding, we performed the same assays using the two truncated mutants. Importantly, although both mutants can be cleaved by the microprocessor, only *pri-mir-19a* truncation v1, which is bound by XPO5 in the absence of RanGTP, was more efficiently cleaved by both the WT and mutant XPO5 (Fig. 3b). In contrast, the processing of *pri-mir-19a* truncation v2, which is not bound by

XPO5, was not further enhanced by XPO5 (Fig. 3c). Together, these data reveal that RanGTP-independent binding of *pri-mir-19a* by XPO5 outside of the pre-miRNA hairpin promotes the microprocessor cleavage in intro.

**XPO5 promotes pri-miRNA processing of clustered miRNAs.** To extend these findings beyond *pri-mir-19a*, we turned to *pri-mir-15b~16-2*, which encodes two miRNAs and also has extensive XPO5 association outside of pre-miRNA hairpin (Supplementary Fig. 1d). Notably, the predicted secondary structure of *pri-mir-15b~16-2* also contains extensive base-pairing regions outside of the pre-miRNA hairpin (Fig. 4a). In support of the HITS-CLIP data, *pri-mir-15b~16-2* bound to XPO5 in a RanGTP-independent manner. Super-shifted bands were also observed with the increasing dose of XPO5 (Fig. 4b), similar to the pattern of *pri-mir-19a*. In addition, an isolated *pri-mir-15b* containing the lower stem region also bound to XPO5 and formed a singly shifted complex with the Kd of 9.57 nM (Fig. 4c–e), similar to the pattern of *pri-mir-19a* truncation v1 (Fig. 2g, h). Together, these in vitro data reveal an unexpected RanGTP-independent binding of XPO5 to *pri-mir-17~92* and *pri-mir-15b~16-2* polycistronic miRNA, *pri-mir-19a* and *pri-mir-15b* precursors.

Because of the RanGTP-independent association of *pri-mir-15b~16-2* and *pri-mir-15b* to XPO5 (Figs. 4b, d), we next tested these primary miRNAs with the processing assay. The processing efficiency of both *pri-mir-15b~16-2* and *pri-mir-15b* by the microprocessor was also strongly enhanced (approximately twofold) by the presence of XPO5 (Fig. 4f, lanes 3–6 and Fig. 4g, lanes 3–5). Notably, the unprocessed *pri-mir-15b~16-2* and *pri-mir-15b* was strongly reduced. We detected both singly cropped *pri-mir-15b~16-2* intermediates and *pre-mir-15b/16-2*, and the accumulation of both was increased by XPO5. In addition, the quadruple XPO5 mutant also promoted the DROSHA/DGCR8 processing (Fig. 4f, lanes 8–11). Taken together, these data suggest that RanGTP-independent binding of *pri-mir-19a, pri-mir-15b~16-2* and *pri-mir-15b* by XPO5 facilitates the DROSHA/DGCR8 processing. These findings reveal an additional function of XPO5 independent of nuclear export.

**XPO5 binds to cellular RNAs with double-stranded regions.** In addition to miRNA precursors, we identified numerous non-coding RNAs in the XPO5 HITS-CLIP datasets (Fig. 5a and Supplementary Data 2). Among these, noncoding RNAs were SINE and LINE elements (repeat), vault RNAs (vtRNAs), 7SL RNA, snRNA, snoRNA, 7SK RNA, tRNA, and hTR (Fig. 5b, g, i and Supplementary Fig. 4). vtRNAs were among the most abundant XPO5-associated RNAs. Notably, vtRNAs were abundantly bound by all core components of miRNA biogenesis and likely gave rise to small RNAs associated with AGO proteins (Fig. 5b). Similar to *pri-mir-19a*, vtRNAs were bound by XPO5 in a RanGTP-independent manner (Fig. 5c). Furthermore, although vtRNAs typically formed one major complex with XPO5, vtRNAs were also super-shifted by the increasing amount of XPO5 protein (Fig. 5c). The Kd of vtRNA and XPO5 association was determined at 36.27 nM (Fig. 4d), weaker than that of *pri-mir-19a* and *pri-mir-15b*. The weaker interaction between vtRNAs and XPO5 was correlated with the less prominent and shorter dsRNA structures of vtRNAs (Fig. 5e and Supplementary Fig. 5a).

To probe the potential effect of XPO5 on the biogenesis and localization of vtRNA, we generated XPO5 knockout (KO) MEF cells (Supplementary Fig. 5b, c, also see below). Interestingly, neither the level nor the cytoplasmic localization pattern of vtRNA was altered in the absence of XPO5 (Fig. 5f). These data suggest that the binding of vtRNA by XPO5 does not affect the maturation, accumulation, and localization of vtRNA.

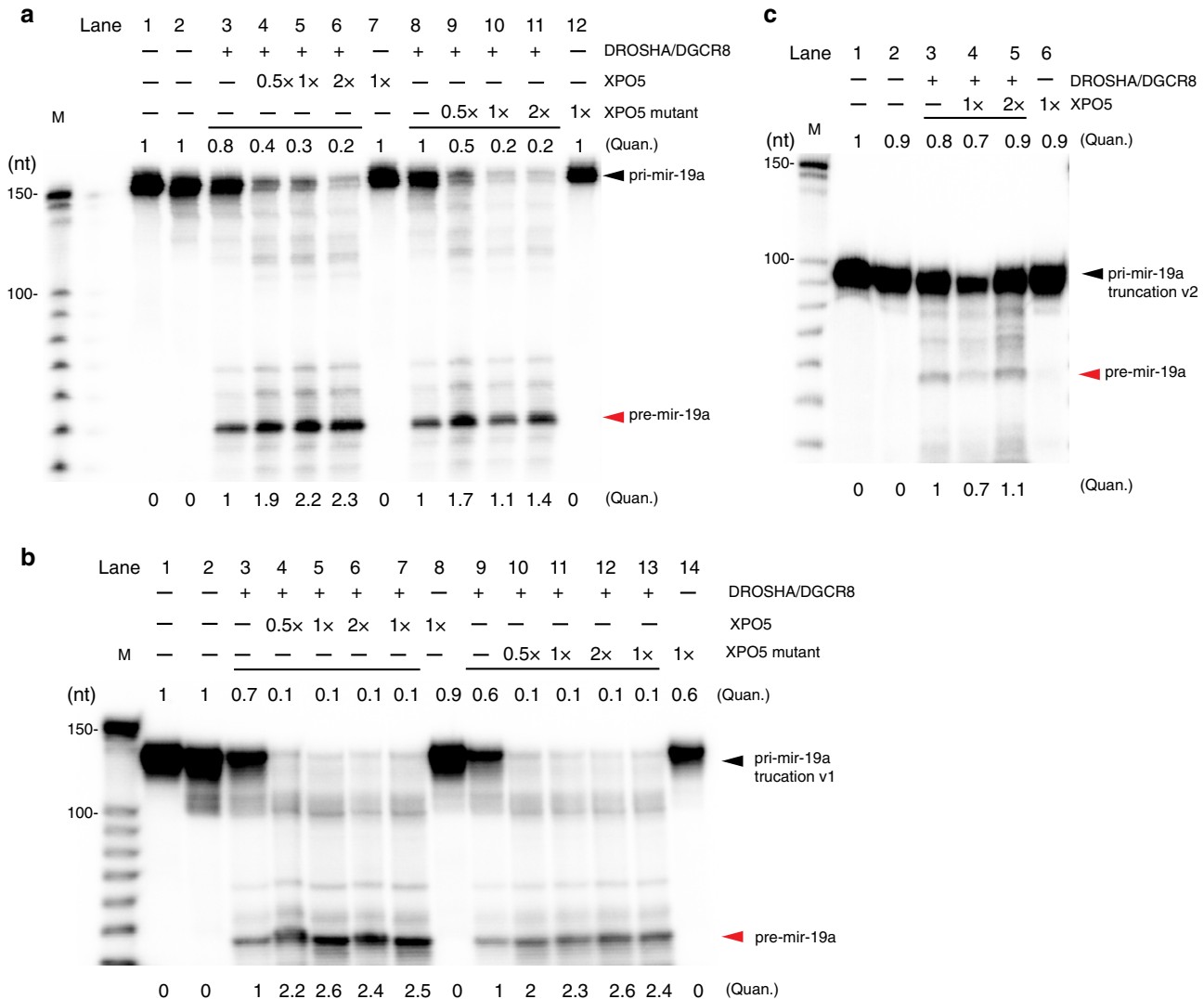

**Fig. 3 XPO5 facilitates the microprocessor cleavage of *pri-mir-19a*. a** Preincubation of XPO5 enhances the microprocessor cleavage of *pri-mir-19a*. Incubation of increasing amount of XPO5 leads to increased processing efficiency of *pri-mir-19a* (lanes 3–6). Incubation of increasing amount of the XPO5 mutant also results in increased processing efficiency of *pri-mir-19a* (lanes 8–11). **b** Preincubation of XPO5 and XPO5 mutant enhances the microprocessor cleavage of truncated *pri-mir-19a* v1. **c** Preincubation of increasing amounts of XPO5 does not promote the cleavage efficiency of *pri-mir-19a* truncation v2 by the microprocessor. Representative images of three independent repeats are shown for each assay (**a**–**c**). (Quan. = quantification). Original data for all panels are provided in the Source Data file. .

We noticed that many of XPO5-associated RNAs are predicted to form double-stranded regions. In general, XPO5 showed more widespread binding patterns than the other core components including DROSHA, DGCR8, and DICER1 in structured RNAs such as hTR and 7SL RNA (Fig. 5g–i). hTR interacts with XPO5 in three domains with the strongest interaction detected towards the 5′ end of hTR. Intriguingly, the number of XPO5 HITS-CLIP reads associated with each domain was different, despite of their origin from the same RNA (Fig. 5g). To probe whether XPO5 association may be correlated with the strength of base-pairing, we further examined XPO5-bound regions on hTR. Three highly structured regions of hTR including the template region, the CR4/CR5 domain and the scaRNA domain have been determined previously[33,34]. Among these regions, the template region forms the most extensive dsRNA structures, followed by the CR4/CR5 and scaRNA domains (Fig. 5h). Notably, XPO5-associated RNA reads were mostly enriched in the double-stranded regions (P2a-P2b-J2b/3) of the template region, followed by the reads over the double-stranded regions of the CR4/CR5 domain and the

scaRNA domain, respectively (Fig. 5g). In addition, AGO interacts with small RNA fragments that are derived from the scaRNA domain located at the 3′ end of hTR but not the more prominent template region or the CR4/5 domain (Fig. 5g). These data suggest that XPO5 preferentially interacts with structured regions of hTR. Similar to hTR, 7SL RNA has well-defined secondary structures with extensive base-pairing regions that are conserved through evolution[35,36]. Interestingly, 7SL RNA strongly interacts only with XPO5 but not with other core components and AGO-associated small RNA fragments were not detected (Fig. 5i), indicating a possible function of XPO5 that is independent of miRNA biogenesis.

To globally measure the preference of XPO5 to base-paired RNAs, we performed an RNA Folding energy analysis for all XPO5-associated RNAs based on HITS-CLIP. We classified these RNAs into repeat associated regions, nonrepeat associated regions and pre-miRNAs. As shown in Fig. 5j, the folding energy of XPO5-associated pre-miRNAs was significantly lower than that of randomly selected sequences and indistinguishable from that

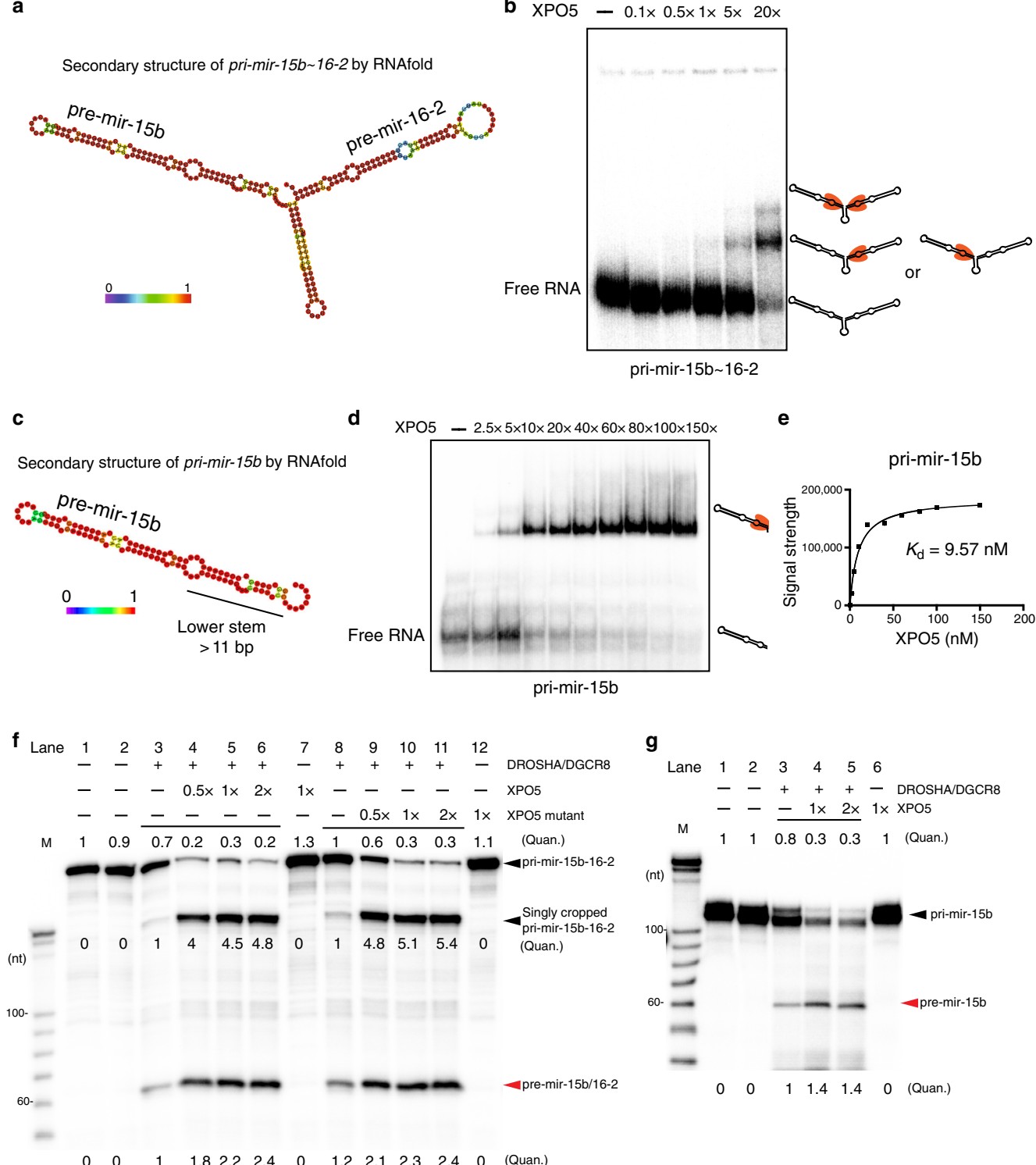

**Fig. 4 XPO5 promotes primary miRNA processing of clustered miRNAs. a** Secondary structure of *pri-mir-15b~16-2* is predicted by *RNAfold*. Heat map indicates the probability of predicted structures (purple: 0—low; red: 1—high). **b** XPO5 binds to *pri-mir-15b-16-2* in a RanGTP-independent manner. Increasing amount of XPO5 results in a super shift. Black hairpin represents *pri-mir-15b~16-2*. XPO5 is coloured in orange. **c** Secondary structure of *pri-mir-15b* is predicted by *RNAfold*. **d** XPO5 binds to *pri-mir-15b* in a RanGTP-independent manner. Black hairpin represents *pri-mir-15b*. XPO5 is coloured in orange. **e** The dissociation constant (Kd) between XPO5 and *pri-miR-15b* is calculated based on the binding results in **d**. **f** Preincubation of XPO5 enhances the microprocessor cleavage of *pri-mir-15b~16-2*. Incubation of increasing amount of XPO5 leads to increased processing efficiency of *pri-mir-15b-16-2* (lanes 3–6). Incubation of increasing amount of the XPO5 mutant also results in increased processing efficiency of *pri-mir-15b~16-2* (lanes 8–11). **g** Preincubation of increasing amounts of XPO5 promotes the cleavage efficiency of *pri-mir-15b* by the microprocessor. Representative images of three independent repeats are shown for each assay (**f**–**g**). (Quan. = quantification). Original data for **b**, **d**, **f**, and **g** are provided in the Source Data file.

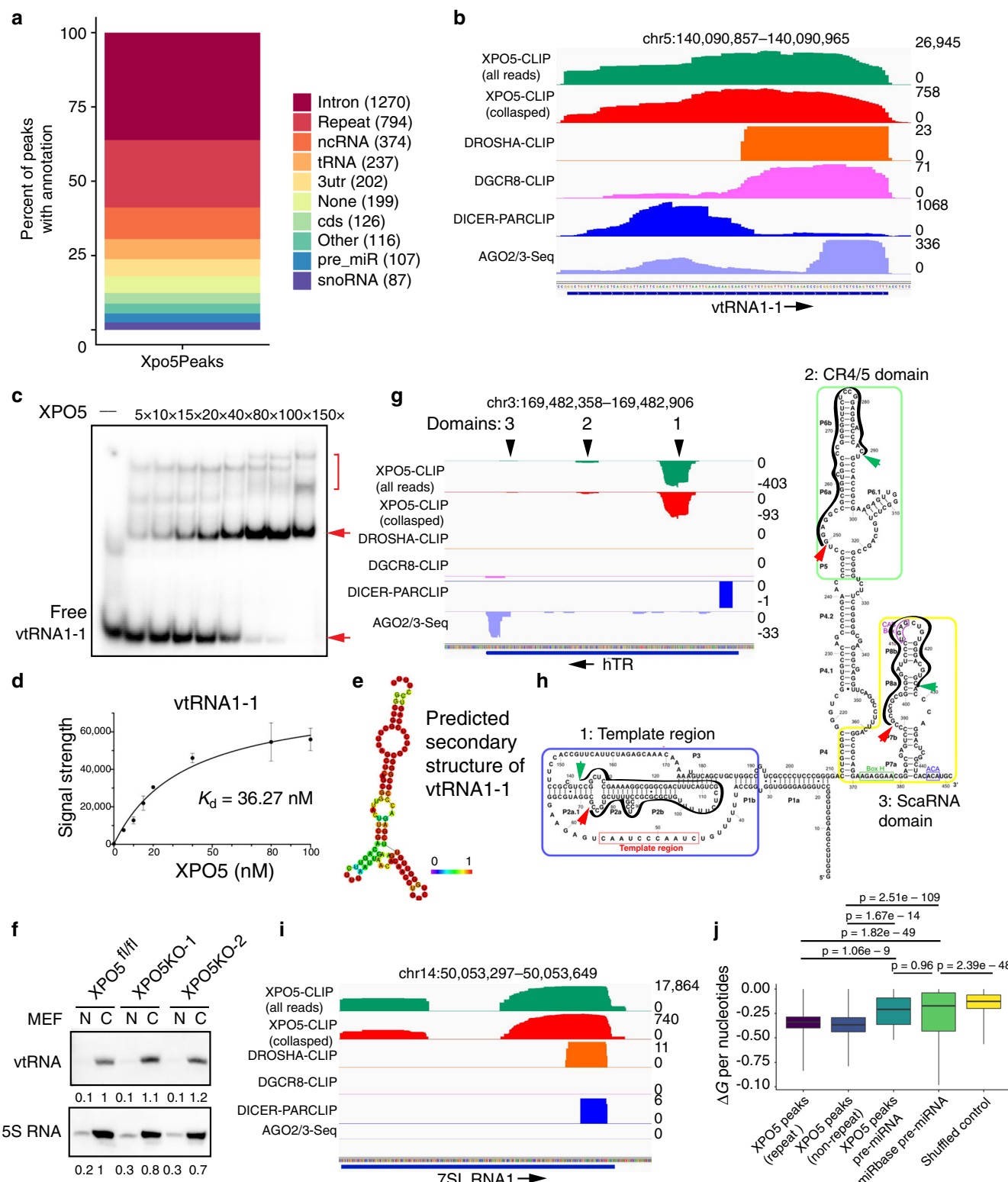

of all pre-miRNA hairpins. Interestingly, both repeated associated and nonrepeat associated regions bound by XPO5 had an even lower folding energy than that of pre-miRNA hairpins (Fig. 5j). These data suggest that XPO5 has a preference to double-stranded regions of cellular RNAs.

**XPO5 is required for mouse embryonic development.** Having characterized XPO5-associated cellular RNAs, we next examined the function of XPO5 in mouse development. We identified a knock-in (KI) first, conditionally targeted embryonic stem cell clone for generating *XPO5* KO mouse models[37]. Founder KI mice were produced and validated for germline transmission of the targeted allele that contains an *IRES-LacZ* and *Neo* cassette in the intron 7 and a floxed exon 8 of *XPO5* (Fig. 6a). To generate the conditional allele, the founder KI mice were bred to *Actb-Flp* mice to remove the *IRES-LacZ* and *Neo* cassette. Finally, the conditional allele was bred to *EIIA-Cre* or *Krt14-Cre* mice to

**Fig. 5 XPO5 associates with diverse cellular RNAs. a** Peak annotation of XPO5-associated reads from XPO5 HITS-CLIP. **b**, **g**, **i** IGV tracks of vtRNA1-1, hTR, and 7SL RNA1 show the association of XPO5 with each noncoding RNA, compared with DROSHA, DGCR8, DICER1, and AGO2/3. Blue bar at the bottom indicates the coding region and the arrow indicates the direction of transcription. Data range is shown on the right of each track. **c** XPO5 binds to vtRNA1-1 without RanGTP. 1 nM vtRNA1-1 substrate was used. Red arrows indicate vtRNA1-1 RNA with (upper) and without (lower) XPO5 binding, red bracket indicates the super-shifted vtRNA1-1 RNA with more than one XPO5 molecule. **d** The dissociation constant (Kd) between XPO5 and vtRNA1-1 is calculated based on the binding results in **c**. **e** Predicted secondary structure of vtRNA1-1. The potential to form predicated structures is coloured (purple to red → low to high). **f** Detection of vtRNA and 5S RNA (loading control) from nuclear and cytoplasmic RNAs extracted from wild type and XPO5 KO MEF cells by northern blotting. **h** XPO5 HITS-CLIP reads align to the secondary structure of hTR. Red and green arrows indicate the 5′ and 3′ location of XPO5-associated RNA regions, respectively. Black lines outline the XPO5-associated hTR regions. **j** An RNA Folding analysis shows the folding energy of XPO5-assoicated repeat RNAs, XPO5-assocatied nonrepeat RNAs, XPO5-associated pre-miRNAs, miRbase pre-miRNAs, and randomly shuttled control sequences. For each boxplot, the middle line is the median, the vertical line spans the data range, and the hinges are the first and third quartiles. A two-sided Mann–Whitney test was used for statistical tests. Original data for **c** and **f** are provided in the Source Data file.

generate constitutive KO and skin cKO of *XPO5*, respectively (Fig. 6a).

Neither *XPO5^{KI/+}* nor *XPO5^{KO/+}* breeding produced any viable *XPO5* null (*XPO5^{KI/KI}* or *XPO5^{KO/KO}*) mice at birth, whereas *XPO5* WT and heterozygous mice were obtained at the expected Mendelian ratio, indicating a requirement of *XPO5* during mouse embryonic development (Fig. 6b). Closer inspection revealed that *XPO5* null embryos showed signs of compromised development around embryonic day 6.5, a developmental stage that is correlated with the onset of gastrulation and embryonic germline layer formation. At this stage, *XPO5* null embryos were much smaller, compared with their control littermates (Fig. 6c). Furthermore, *XPO5* null embryos failed to go through gastrulation and initiate organogenesis by E8.5 (Fig. 6d). Judging by the LacZ signals in E7 *XPO5^{KI/+}* and *XPO5^{KI/KI}* embryos, *XPO5* was universally expressed in all embryonic cells (Fig. 6e).

We then performed quantitative real-time PCR (qRT-PCR) analyses to validate the loss of *XPO5* and *mir-290*, a representative miRNA that is highly expressed in mouse ESCs and early embryos, using total RNA isolated from E7 WT, het, and KO embryos (Fig. 6f). Because the miRNA pathway is required to promote the differentiation of ESCs to lineage specific cell types and the loss of *Dicer1* and *Dgcr8* blocks ESC differentiation[38,39], we also quantified ESC marker genes such as *Nanog* and *Oct4* as well as differentiation markers such as *T* and *Tet1* by qRT-PCR. The results revealed a strong accumulation of *Nanog* and *Oct4* mRNAs in the E7 *XPO5* null embryos, and a relatively normal level of *T* and *Tet1* (Fig. 6g). Furthermore, the high level of *Oct4* persisted in E8.5 *XPO5* null embryos (Fig. 6h), reflecting a compromised differentiation. Together, these data reveal that *XPO5* is required for miRNA biogenesis and mouse embryonic development.

**XPO5 is required for skin morphogenesis**. We have previously studied *Dicer1*, *Dgcr8*, as well as *Ago1/2* in the skin using a *K14-Cre* line and observed unique defects in hair morphogenesis that are characteristic of the loss of the entire miRNA pathway in the skin[12,18,19]. Furthermore, both *Dicer1* and *Dgcr8* cKO skin typically loses >95% of individual miRNAs with a few exceptions of short hairpin miRNAs such as *mir-320* and *mir-484* that are dependent on *Dicer1* but independent of *Dgcr8*[19]. In comparison, *Ago1/2* dKO skin loses ~80% of individual miRNAs and shows milder defects[12]. To compare the developmental defects of *XPO5* cKO animals with these well-characterized cKO models of the miRNA biogenesis pathway in the skin, we generated *K14-Cre/ XPO5^{fl/fl}* mice (Fig. 7a).

We first documented the expression pattern of *XPO5* in the skin by monitoring LacZ signals in *XPO5^{KI/+}* heterozygous mice. *XPO5* was universally expressed in all skin cell types at postnatal day 2 and 4 (P2 and P4). By P4, when hair follicles gave rise to

terminally differentiated inner root sheath and hair shaft, *LacZ* signals appeared to be elevated in these differentiated cells than their progenitor counterparts (Fig. 7b). In *XPO5* skin cKO animals, we observed progressively compromised hair follicle development that was similar but slightly milder, compared with *Dicer1* and *Dgcr8* cKO (Fig. 7c). *XPO5* cKO animals showed neonatal lethality between postnatal day 5 and 18 and failed to gain weight postnatally (Fig. 7c–e). This was in contrast to *Ago1/2* dKO animals, which lose ~80% of total miRNAs and usually survive to adulthood despite the apparent loss of hair follicles[12]. Consistent with these observations, qRT-PCR quantification of *mir-203* and *mir-205*, two of the most highly expressed miRNAs in the skin, showed >90% depletion for both miRNAs (Fig. 7f). In addition, AGO2 protein was also depleted in *XPO5* null skin (Fig. 7g). Because mature miRNA accumulation is required to stabilize AGO proteins[40], this result provided further support to the strong depletion of global miRNA expression in the absence of *XPO5*.

To further document the role of *XPO5* during skin morphogenesis, we examined neonatal skin. We observed reduced hair follicle formation and stunted hair follicle down growth in P2 skin (Fig. 7h). When examined specifically with *Lef1* and *β4-integrin* staining for hair germ, we found many evaginating hair germs towards the epidermis that is characteristic of *Dicer1* and *Dgcr8* cKO skin[17–19] (Fig. 7i). Furthermore, epidermal cell proliferation was strongly reduced (Fig. 7j). Finally, epidermal differentiation was not strongly affected as documented by the *Krt5* and *Krt1* staining that illuminated the basal and differentiated spinous layer, respectively (Fig. 7k) and by qRT-PCR analysis (Fig. 7l). Collectively, these studies provide further genetic evidence to support the requirement of *XPO5* for miRNA biogenesis and skin development in mouse.

**XPO5 is required for miRNA biogenesis**. Having demonstrated the requirement of *XPO5* in mouse embryonic and tissue development, we further characterized the global miRNA expression levels in *XPO5* KO skin using a quantitative small RNA sequencing method[41] (Fig. 8a). Globally, small RNAs between 20 and 23 nt showed a strong depletion in the *XPO5* KO samples (Fig. 8b). We then specifically examined miRNA depletion. Although it was somewhat weaker than the depletion observed in the *Dicer1* and *Dgcr8* cKO skin (Fig. 8c), the average level of depletion was ~90%. The depletion of mature miRNA was particularly evident for highly expressed miRNAs whose expression can be more robustly measured by small RNA sequencing (Fig. 8d and Supplementary Data 3). In addition to highly expressed *mir-203* and *mir-205* (Fig. 7f), we quantified the depletion of five miRNAs from the *mir-17~92* and *mir-15~16* clusters (Fig. 8e, f), which expressed at an intermediate level in the skin. We observed strong depletion for *mir-17-5p*, *mir-18*, *mir-19a/b*, and *mir-20a* except *mir-92a*

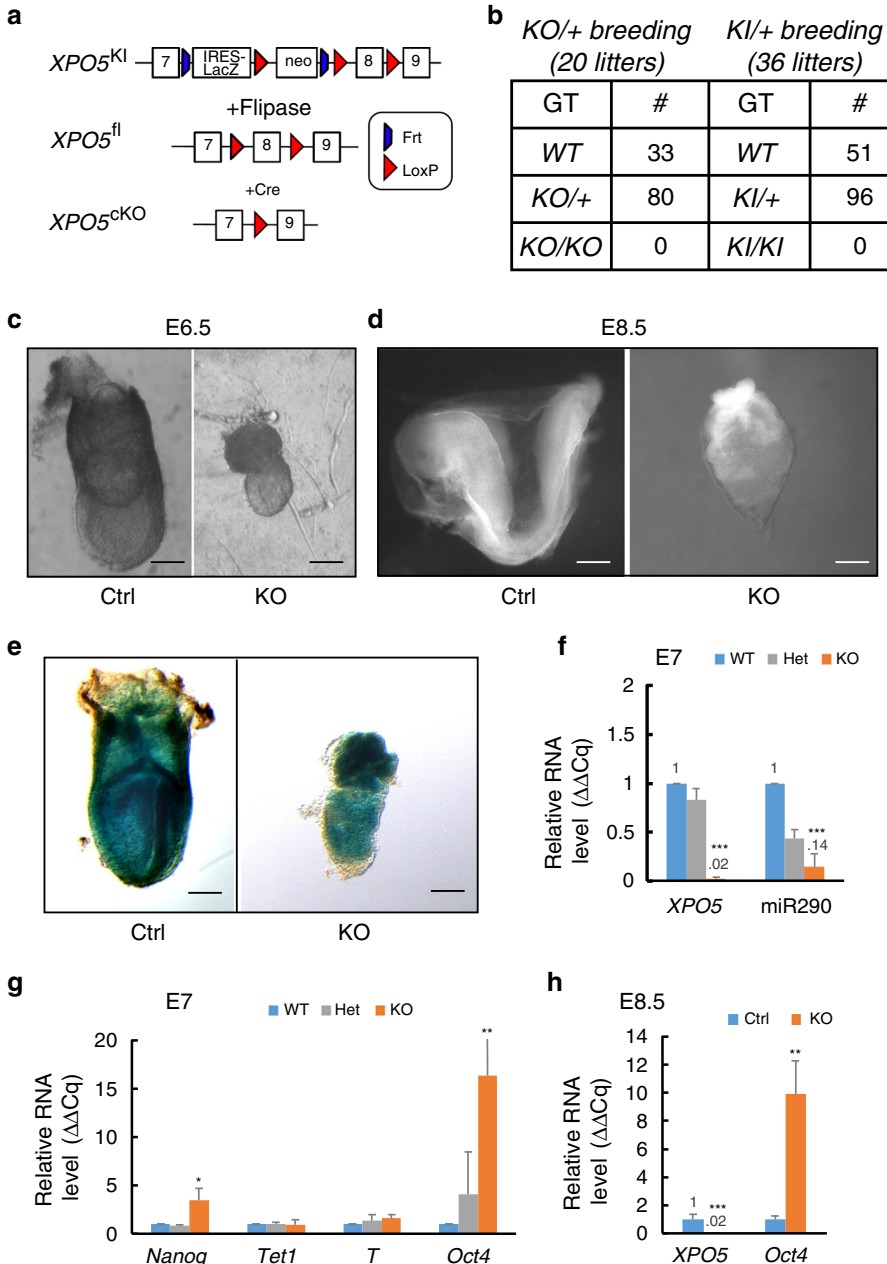

**Fig. 6 XPO5 is required for mouse embryonic development. a** Schematics of the *XPO5* allele design. **b** Loss of *XPO5* results in embryonic lethality. *XPO5* KO embryos show developmental defects as early as in E6.5 (**c**) and fails to go through gastrulation (**d**). Representative images of littermate embryos are shown for over ten control and KO embryos. **e** Whole-mount *LacZ* staining of *XPO5^KI/+* (Ctrl) and *XPO5^KI/KI* (KO) shows universal expression of *XPO5* throughout the embryo. **f** Depletion of *XPO5* and *mir-290* in *XPO5* KO embryos at E7 is measured by qPCR. Dysregulation of selected genes in *XPO5* KO embryos at E7 (**g**) and E8.5 (**h**) is measured by qPCR, respectively. *Oct4* and *Nanog* are failed to be downregulated in *XPO5* KO embryos at both E7 and E8.5. Scale bar 250 μm in **c–e**. Data shown are mean s.d. from three independent experiments. *P < 0.05; **P < 0.01; ***P < 0.001 determined by Student's *t* test. Original data for **f**, **g**, and **h** are provided in the Source Data file.

as well as *mir-15a/b* and *mir-16* (Fig. 8e, f). Interestingly, the loss of *mir-19a/b* was particularly strong, likely reflecting the requirement of XPO5 for its processing and nuclear export. To further validate the compromised miRNA biogenesis and distinguish the effect of XPO5 loss on different miRNA species, we performed northern blotting for *mir-17* (Fig. 8g). We observed the loss of mature *mir-17* but no strong changes of *pre-mir-17* were detected (Fig. 8g). The lack of accumulation of pre-miRNAs is likely due to their unstable nature in the absence of XPO5, similar to the patterns reported in *Dicer1* cKO[18]. In addition, *mir-320* and *mir-484*, whose biogenesis is independent of *Drosha/Dgcr8* processing[19], were largely unchanged in *XPO5* cKO skin (Fig. 8h), lending support to the notion that *XPO5* is required to export pre-miRNA hairpin generated by the *Drosha/Dgcr8* microprocessor. Taken together, these analyses reveal the requirement of *XPO5* for the biogenesis of a majority of miRNAs in the skin, consistent with the strong defects observed in the cKO model. We also note, however, whether the phenotypes of XPO5 KO is entirely due to the loss of miRNAs should be further examined due to the binding of XPO5 to many cellular RNAs with double-stranded regions.

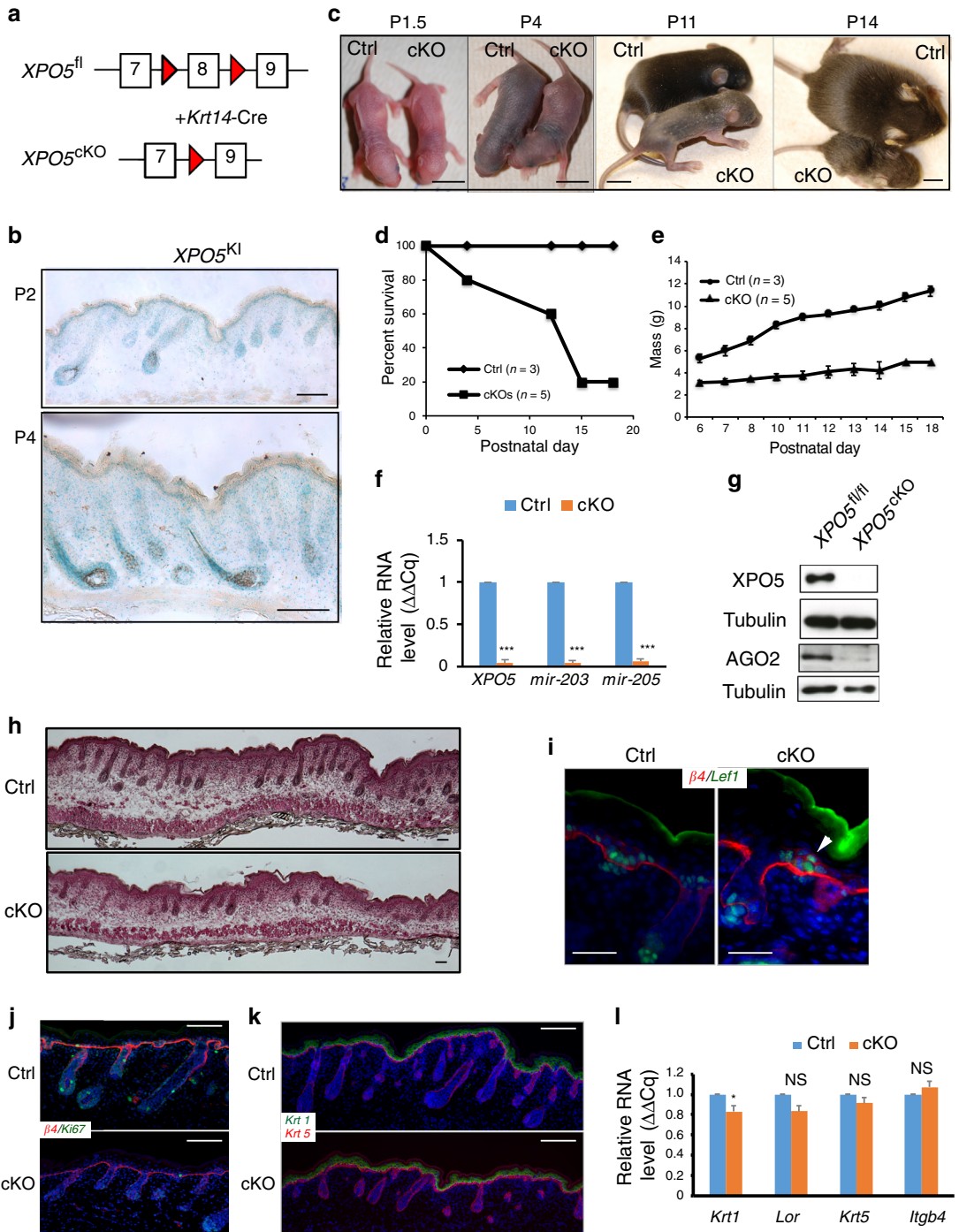

**Fig. 7 XPO5 is required for skin development. a** Schematic of *K14-Cre/XPO5*$^{fl/fl}$ mouse model. **b** Whole-mount *LacZ* staining of *XPO5*$^{KI/+}$ in P2 and P4 skin. **c** Neonatal mice of *XPO5* cKO animals with littermate controls from P1.5 to P14. **d, e** *XPO5* cKO animals are neonatal lethal and fail to gain weight. **f** Depletion of *mir-203* and *mir-205* in *XPO5* cKO epidermis is quantified by qPCR. Data shown are mean s.d. from three independent experiments. ***$P < 0.001$ by Student's *t* test. **g** Depletion of XPO5 and AGO2 proteins in *XPO5* cKO epidermis is confirmed by western blot. **h** H&E staining reveals reduced hair follicle formation and stunted hair follicle growth in *XPO5* cKO skin at P2. **i** Stunted hair follicle down growth and evaginating hair germ (arrowhead) in *XPO5* cKO skin are revealed by *Lef1* and *β4-integrin* staining. **j** Reduced cell proliferation in *XPO5* cKO skin is shown by Ki67 staining. **k** *XPO5* cKO skin shows normal epidermal differentiation as determined by Krt5 and Krt1 staining. **l** Normal epidermal differentiation is confirmed by qPCR detection of *Krt1, Lor, Krt5*, and *Itgb4* (encodes β4-integrin). Data shown are mean s.d. from three independent experiments. *$P < 0.05$, n.s.—not significant, by Student's *t* test. Scale bar: 1 cm in **c**, 100 μm in **b, h, j**, and **k**, 50 μm in **i**. Original data for **d–g** and **l** are provided in the Source Data file.

## Discussion

In this study, we have determined XPO5-associated RNA species at the genomic scale. As expected, XPO5 associates with pre-miRNA precursors for a vast majority of expressed miRNAs (Fig. 1b). Surprisingly, however, XPO5 strongly binds to primary miRNA precursors of some closely clustered polycistronic miRNAs such as *mir-17~92* and *mir-15b~16-2* (Fig. 1i–k and Supplementary Fig. 1). The in vitro binding

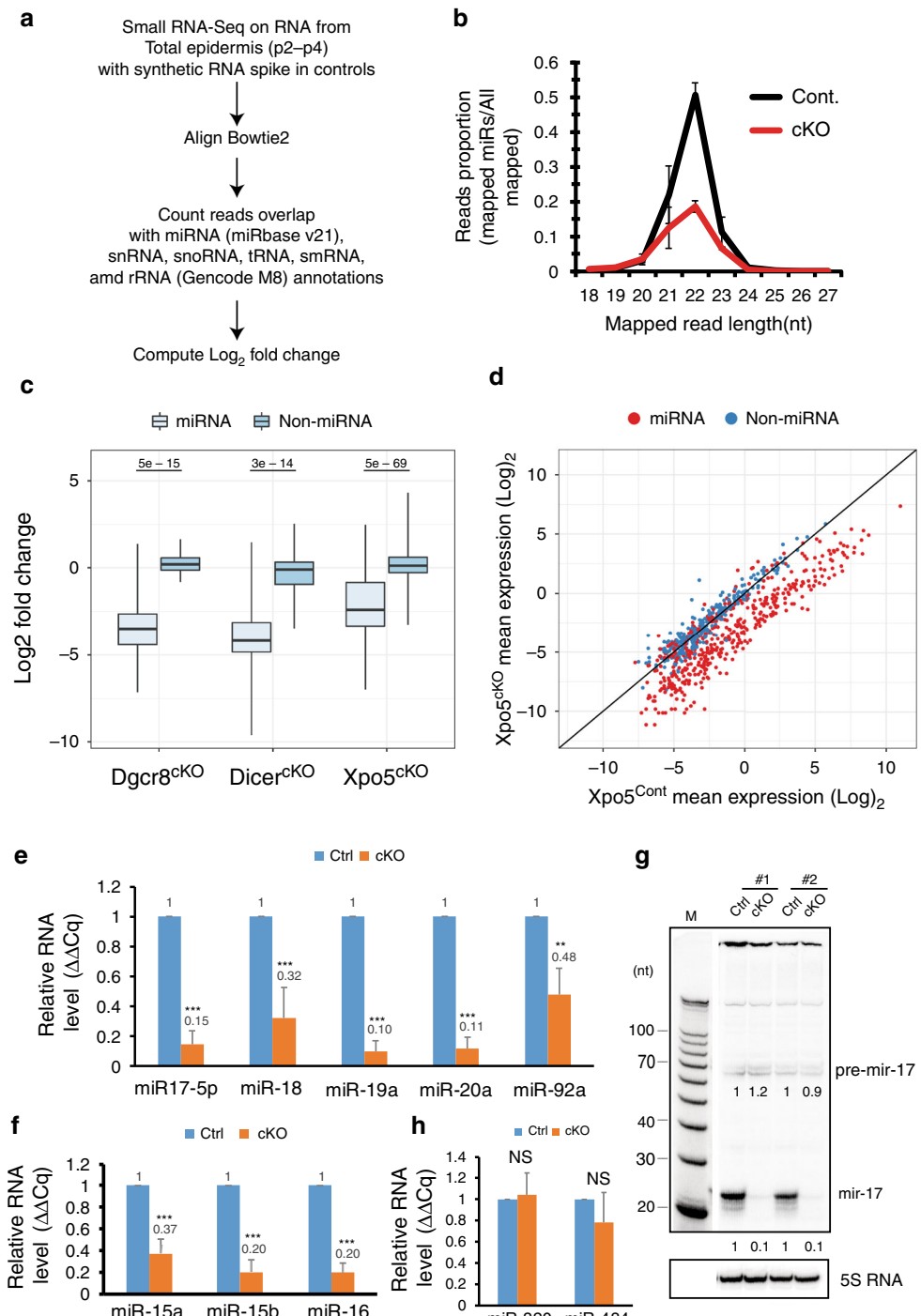

**Fig. 8 *XPO5* is required for miRNA biogenesis. a** Bioinformatic pipeline of quantitative small RNA-seq and data analysis. **b** Small RNAs between 20 and 23 nt showed strong depletion in *XPO5* cKO skin samples. Small RNA reads from 18 to 27 nt were charted from small RNA cDNA libraries. **c** The depletion of miRNAs in *XPO5* cKO skin is weaker than that in *Dicer1* and *Dgcr8* cKO skin. For each boxplot, the middle line is the median, the vertical line spans the data range, and the hinges are the first and third quartiles. A two-sided Mann–Whitney test was used for statistical tests. **d** Depletion of mature miRNA reads in *XPO5* cKO skin is evident for highly expressed miRNAs (red coloured dots). **e** Depletion of *mir-17-5p*, *mir-18*, *mir-19a*, and *mir-20a* except *mir-92a* in *XPO5* cKO skin is measured by qPCR. **f** Depletion of *mir-15a*, *mir-15b*, and *mir-16* in *XPO5* cKO skin is measured by qPCR. **g** Depletion of *mir-17-5p* is confirmed by northern blotting. **h** Unchanged expression of *mir-320* and *mir-484* in *XPO5* cKO skin is measured by qPCR. Data shown are mean s.d. from three independent experiments. **P < 0.01; ***P < 0.001, n.s.—not significant by Student's *t* test. Original data for **e–h** are provided in the Source Data file.

assays provided evidence that XPO5 binds to these unprocessed primary miRNAs in a RanGTP-independent manner (Figs. 2 and 4). Studies of the *mir-19a* precursors further suggest that the binding site of XPO5 to these primary miRNAs is in the

extensive base-pairing regions beyond pre-miRNA hairpin. Because RanGTP-dependent cargo association is characteristic of exportin-mediated nuclear export[1], the RanGTP-independent association with these primary miRNAs suggests

a hitherto unrecognized function of XPO5. Indeed, incubation of XPO5 with both *pri-mir-19a, pri-mir-15b~16-2*, and *pri-mir-15b* increased the processing efficiency of the DROSHA/DGCR8 microprocessor, providing evidence for the involvement of XPO5 in the nuclear cleavage of closely clustered, polycistronic miRNAs. Given the lack of association of XPO5 with monocistronic or sparsely clustered miRNAs, it is possible that the XPO5-mediated mechanism for efficient pri-miRNA cleavage is unique to closely clustered miRNAs with extensive dsRNA regions. It is tempting to speculate that closely clustered miRNAs may use multiple mechanisms including XPO5 binding to regulate their biogenesis, conferring more complex control to the production of these miRNAs post-transcriptionally. However, how many of these closely clustered miRNAs whose nuclear cleavage is regulated by XPO5 should be investigated in future studies in a cell type-specific manner.

Second, XPO5 pervasively associates with many cellular RNAs in addition to miRNA precursors. Global RNA folding analyses and individual case examination suggest that XPO5 has a preference to dsRNA regions. In addition to miRNA precursors, XPO5 binds numerous cellular RNAs such as vtRNA, 7SL RNA, snRNA, snoRNA, 7SK RNA, tRNA, hTR, and SINE and LINE repeat elements, which have diverse cellular localization and functions[42]. Interestingly, the Kd of XPO5 to *pri-mir-19a* variants and vtRNA is similar to the Kd of MDA5 and dsRNA[43]. Although functional consequences of these binding events will be examined in future studies, it is possible that *XPO5* has multiple functions beyond miRNA biogenesis[1]. We note mammalian XPO5 was originally identified as a binding protein to dsRNA binding proteins such as ILF3, PKR, and STAUFEN[16]. Our results now provide a molecular basis at the genomic scale for these earlier findings and establish a foundation to investigate the link between XPO5 and cellular dsRNAs and their binding proteins.

Finally, we have unequivocally demonstrated the requirement of *XPO5* for global miRNA biogenesis and mouse development by examining *XPO5* KO in embryonic and skin development. In both systems examined, we observed strong developmental defects reminiscent of those observed in *Dicer1* and *Dgcr8* KO animals. In the skin, we have previously shown that the loss of *Dicer1* or *Dgcr8* leads to complete depletion of most miRNAs with a couple of exceptions such as *mir-320* and *mir-484*, which are *Dicer1*-dependent but *Dgcr8*-independent[18,19]. These cKO animals were neonatal lethal and died before one-week old. In contrast, KO of *Ago1* and *Ago2* together (dKO) leads to 70–80% depletion of most miRNAs due to quantitative loss of AGO proteins and the dKO mice usually survived to adulthood with prominent defects in hair follicles[12]. Collectively, these models set useful boundaries to estimate the requirement of *XPO5* for miRNA biogenesis if miRNAs are not completely depleted in the absence of *XPO5*. Our miR-seq, qPCR and northern blotting analyses suggested that KO of *XPO5* leads to ~90% depletion of most miRNAs. Of note, *Dgcr8*-independent miRNAs, *mir-320* and *mir-484*, were not affected by the loss of *XPO5*. This result is consistent with a previous report for *XPO5*-independent miRNA export[9] and indicates an intrinsic link between DROSHA/DGCR8 produced pre-miRNA hairpin and *XPO5*-mediated export. In support of these molecular measurements, *XPO5* skin cKO animals lived longer than their *Dicer1* or *Dgcr8* cKO counterparts but shorter than Ago1/2 dKO animals. Although other factors could weakly compensate for the loss of *XPO5* for miRNA expression, they are unable to rescue the developmental defects. Therefore, we conclude that *XPO5* is broadly required for miRNA biogenesis and mouse development. We note, however, because XPO5 also binds to many non-miRNA substrates the phenotypes of XPO5

KO may not be entirely due to the loss of miRNA biogenesis. This possibility warrants future investigations.

## Methods

**Mice**. All WT and transgenic animal breeding and operation procedures were approved by the Institutional Animal Care and Use Committees (IACUC) at the University of Colorado Boulder and in accordance with the guidelines and regulations for the care and use of laboratory animals, and compiled with all relevant ethical regulations for animal testing and research, and received relevant ethical approvals. The *XPO5*KI animal was purchased as an ES cell clone from the EU's KO mouse consortium. *EIIa-Cre* mice were obtained from the Jackson laboratory (JAX #003724). The mouse was crossed with Flipase expressing mice to generate the *XPO5*fl/fl mice. *XPO5*KI, *XPO5*fl/fl, and K14-Cre; *XPO5*fl/fl mice were bred and housed in the University of Colorado Boulder. Embryos studies were timed by the presence of a plug, indicating gestational age E0.5. All mouse experiments were conducted in accordance with animal protocols approved by the IACUC.

**Cell culture**. HEK293T cells, obtained from ATCC (ATCC CRL-3216), were cultured and maintained in DMEM (GIBCO) supplemented with 10% heat-inactivated fetal bovine serum and 1% penicillin/streptomycin (GIBCO) in the 5% $CO_2$ incubator at 37 °C. *XPO5*fl/fl MEF cells were isolated from E14 *XPO5*fl/fl mice using the Pierce™ Mouse Embryonic Fibroblast Isolation Kit (#88279, Thermo Fisher Scientific). *XPO5* KO MEF cells were generated by Adeno-cre virus infection. *XPO5*fl/fl and *XPO5* KO MEF cells were cultured and maintained in DMEM for Pierce™ Primary Cell Isolation Kits (#88287, Thermo Fisher Scientific) supplemented with 10% heat-inactivated fetal bovine serum and 1% penicillin/streptomycin (GIBCO) in the 5% $CO_2$ incubator at 37 °C.

**RNA purification and mRNA and miRNA qPCR**. Total RNA was extracted using Trizol (Thermo Fisher Scientific). For mRNA analysis, 1 µg total RNA was used to synthesize cDNA by Superscript III Reverse Transcriptase (Thermo Fisher Scientific). For miRNA analysis, the miScript II RT Kit (Qiagen) was used to synthesize cDNA. Reactions were performed according to the manufacturer's manual and on a CFX384 real-time system (Bio-Rad). Differences between samples and controls were calculated using the $2^{-\Delta\Delta C(t)}$ method. All the primers used for qPCR are listed in Supplementary Table 1.

**Immunostaining and images**. For analysis of back skin phenotypes, OCT sections were fixed in 4% PFA for 10 min in phosphate-buffered saline (PBS) and washed three times for 5 min in PBS at room temperature. Block the sections with 2.5% NGS, 2.5% NDS in PBS. Primary antibodies against the following proteins were used: β4-integrin (β4, 1:100, BD Biosciences), Lef1 (1:500, Cell signaling), K5 (1:5000, Covance), K1 (1:2000, Covance), and Ki67 (1:500; Abcam). Imaging was performed on a Leica DM5500B microscope with an attached Hamamatsu C10600-10B camera and MetaMorph (version 7.7; MDS Analytical Technologies) software. For all images, single optical sections were used.

**X-gal histochemistry staining**. Thick sections (25 µm) of back skin samples were fixed in fix solution (0.5% glutaraldehyde, 1.25 mM EGTA pH 7.3, 2 mM $MgCl_2$, 1xPBS) for 10 min at room temperature. For the embryos, whole-mount embryos were fixed in fix solution for 5 min at room temperature. Fixed sections and embryos were washed twice with detergent rinse (100 mM phosphate buffer pH 7.4, 2 mM $MgCl_2$, 0.01% sodium deoxycholate, 0.02% NP-40), and then incubated in staining solution (100 mM phosphate buffer pH 7.4, 2 mM $MgCl_2$, 0.02% NP-40, 5 mM potassium ferricyanide, 5 mM potassium ferrocyanide, 20 mM Tris pH 7.5, 1 mg/mL X-gal) at 37 °C for 48 h.

**HITS-CLIP**. XPO5 HITS-CLIP and data analyses were performed as described[44] with slight modifications[41]. In Brief, HEK293T cells were UV-crosslinked, resuspended in lysis buffer. Supernatants were incubated with Anti-Exportin-5 antibody (Abcam) or control IgG for 1 h on ice with or without RNase. After incubation, dynabeads were added into the protein–RNA complexes solutions and incubate for 1 h at 4 °C, followed by labeling 5′ end of RNA with [γ-32P] ATP using PNK. Then 5′ Ligation Adapter-NN was linked to 5′ end of RNA using T4 RNA ligase 1 at room temperature for 2 h. Protein–RNA complexes were incubated in 1x Nu-PAGE loading buffer (Invitrogen) at 70 °C for 10 min, resolved on 8% Nu-PAGE Bis-Tris gel (Invitrogen), transferred into a nitrocellulose membrane and exposed to a Fuji film. Bands corresponding to specific protein–RNA complexes were excised, and then digested with proteinase K. RNAs were purified and adenylated linker was linked to 3′ end of purified RNAs. cDNAs were made using Superscript III(Invitrogen) with 3′ adenylated linker primers. cDNAs were separated on 15% denaturing urea polyacrylamide gel and one area around 100 bp were excised purified, followed by amplification by Phusion High Fidelity polymerase (NEB). PCR products were separated by 6% denaturing urea polyacrylamide gel and three bands around 100–150 were purified. Obtained CLIP libraries were subjected to high-throughput deep sequencing.

**Small RNA-seq**. Total RNA samples from P2–P4 back skin epidermis were subject to high efficiency 3′ ligation as described previously[41]. Products were resolved on 15% PAGE-urea gels and stained with SybrGold (Molecular Probes S-11494 Carlsbad, CA, USA). The region corresponding to ligated miRNAs (46 nt) was excised, gel slices were thoroughly minced, and eluted in HSCB (400 mM NaCl, 25 mM Tris-HCl pH 7.5, 0.1% SDS) overnight at 4 °C. Nucleic acids were precipitated in the presence of glycogen carrier via 0.1 volumes sodium acetate and 2.5 volumes ethanol. Pellets were washed in 70% ethanol, and dissolved in the 5′ ligation mix without enzyme. Samples were briefly heated to 70 °C, snap chilled on ice, and then enzyme was added. Following ligation reactions cDNA was prepared using Superscript III RT (Invitrogen) according to the manufacturer's recommendation using 3′ linker-specific RT primer. cDNA libraries served as templates for PCR amplification; amplicons were resolved on 8% native acrylamide gels. Bands of the correct molecular weight were isolated and used for high-throughput sequencing on the Illumina HiSeq2000 and HiSeq4000.

**Plasmids construction and transfection**. Constructs coding for FLAG-tagged human XPO5, XPO5 mutant and RanQ69L were cloned in the pcDNA3.1 vector for transient expression. For the mir-17~92a pri-miRNA in vitro transcription plasmid pJMC101-T7-pri-miR17~92a, human mir-17~92a pri-miRNA was cloned using mir-17~92a pri-miRNA transcription forward and reverse primers (including T7 promotor) and T7:pri-mir-17~92a were inserted into pJMC101vector. Primers for plasmid construction are listed in Supplementary Table 1. HEK293FT transfection was performed using Mirus TransIT®-LT1 Transfection Reagent (# MIR 2300, Fisher) according to the manufacturer's instructions.

**Protein expression and purification**. For XPO5-His$_6$ and His$_6$-RanQ69L proteins expression, pQE60-XPO5 and pQE32-RanQ69L plasmids are transformed separately into E.coli strain TG1 cells[16]. The XPO5 subcloned cells were grown in LB medium with 2% ethanol (vol/vol) at 37 °C, followed by induction with 400 μM IPTG at 18 °C for 16 h. XPO5-His$_6$ was purified on NTA-Ni2 beads (QIAGEN). For His6-RanQ69L protein, subcloned cells were grown in LB medium at 37 °C, followed by induction with 100 μM IPTG at room temperature overnight. His$_6$-RanQ69L was purified on NTA-Ni2 beads and loaded GTP on ice for 3 h[29].

**Electrophoretic mobility shift assay**. Cold mir-17~92a pri-miRNA, pre-mir-30a, pre-mir-19a, pri-mir-19a, pri-mir-19a truncation v1, pri-mir-19a truncation v2, pri-mir-15b~16-2, pri-mir-15b, and vtRNA1-1 transcripts were produced by the MEGAshortscript™ T7 Transcription Kit (#AM1354, Thermo Fisher Scientific) using pJMC101-T7-pri-mir-17~92 template (digested by HindIII), T7-pre-mir-30a, T7-pre-mir-19a, T7-pri-mir-19a, T7-pri-mir-19a truncation v1, T7-pri-mir-19a truncation v2, T7-pri-mir-15b~16-2, T7-pri-mir-15b, and T7-vtRNA1-1 PCR products separately as the templates (primers are listed in Supplementary Table 1), and purified by 5% PAGE gel. The 5′-phosphate of those cold substrates were removed by Antarctic Phosphatase (# M0289S, NEB). After dephosphorylation, those substrates were end-labeled by [r-32P]-ATP using T4 Polynucleotide Kinase (# M0201S, NEB). miR17~92a, miR19a, and miR15b-16-2 pri-miRNAs were refolded in refolding buffer {1XTHE (66 mM HEPES, 33 mM Tris, 1 mM EDTA), 100 mM NH$_4$OAc, 5 mM MgOAc, 0.5% NP-40, 0.1 mM EDTA} at 60 °C for 10 min and slowly cooled down to 25 °C in 25 min. After refolding, mir-17~92a, mir-19a, mir-15b, mir-15b~16-2 pri-miRNAs, and vtRNA1-1 were separately incubated with XPO5 protein in a total volume of 10 μl at 25 °C for 45 min. Pre-mir-30a and pre-mir-19a were incubated with XPO5 protein and RanQ69L. The binding buffer contained 20 mM HEPES (pH 7.3), 150 mM potassium acetate, 2 mM magnesium acetate, 0.05% NP-40, 7 mM 2-mercaptoethanol, 1.5 μg/mL poly dIdC, and 0.2% BSA. After incubation, 0.3 μg of heparin was added to the reaction and incubated for another 5 min[4]. Samples were then analyzed by electrophoresis on a 0.7% nondenaturing agarose gel (for mir-17~92a pri-miRNA) at 4 °C or 5% nondenaturing PAGE gel (for pre-mir-30a, pre-mir-19a, pri-mir-19a, pri-mir-15b, mir-15b~16-2 pri-miRNAs, and vtRNA1-1) at room temperature. Gels were dried and detected by phosphorimager.

**In vitro analysis of pri-miRNA processing**. [α-32P]-CTP (#BLU008H250Uc, Perkin) labeled miR19a, miR15b, and miR15b-16-2 pri-miRNA transcripts were produced by the MEGAshortscript™ T7 Transcription Kit (#AM1354, Thermo Fisher Scientific) using T7 promotor-driven pri-MIRNA gene PCR products as the templates, and purified by 5% PAGE gel. pCK-Drosha-FLAG and pCK-FLAG-Dgcr8, pCDNA3-FLAG-XPO5, pCDNA3-FLAG-XPO5_mutant, and pCDNA3.1 plasmids were separately transfected into Human 293FT cells. After 48 h of culturing, cells were harvested and opened by hypotonic gentle lysis buffer (10 mM Tris-HCl pH 7.6, 10 mM NaCl, 2 mM EDTA, 0.5% Triton X-100, proteinase inhibitor). DROSHA and DGCR8 complex, XPO5 protein, and XPO5 mutant protein were purified by ANTI-FLAG®M2 Affinity Gel (#A2220, Sigma). XPO5 and XPO5 mutant proteins were eluted from ANTI-FLAG® M2 Affinity Gel by adding excess 3X FLAG® Peptide (#F4799, Millipore Sigma). [α-32P]-CTP labeled miR15b-16-2, miR15b, and miR19a pri-miRNA substrates were refolded in refolding buffer (1XTHE (66 mM HEPES, 33 mM Tris, 1 mM EDTA), 100 mM NH40Ac, 5 mM MgOAc, 0.5% NP-40, 0.1 mM EDTA) at 60 °C for 10 min and slowly cooled down to 25 °C in 25 min. After refolding, miR15b-16-2, miR15b, and

miR19a pri-miRNA substrates were incubated with XPO5 or XPO5 mutant protein at 25 °C for 25 min. Then, DROSHA and DGCR8 complex was added into system and incubated at 37 °C for 1 h in processing buffer (100 mM Tris-HCl pH 7.6, 500 mM KCl, 1 mM EDTA, 64 mM MgCl$_2$). Processing products were purified by phenol/chloroform and run on the 5% PAGE denaturing gel. Gels were dried and detected by phosphorimager.

**RNA northern blot**. Nuclear and cytoplasmic RNAs were extracted from MEF cells using the Cytoplasmic & Nuclear RNA Purification Kit (# 21000, Norgen Biotek). Three micrograms of cytoplasmic RNA and two micrograms of nuclear RNA were separated by electrophoresis on 5% PAGE gels and electrically transferred into nylon N+ membrane. [γ-32P] ATP-labeled specific oligonucleotide probe sequences (vtRNA and 5S) were used and hybridized at 42 °C overnight. Total RNA was extracted from P4 back skin epidermis using Trizol (# 15596-018, Thermo Fisher Scientific). five micrograms of total RNA was separated by electrophoresis on 5% PAGE gels and electrically transferred into nylon N+ membrane. [γ-32P] ATP-labelled specific oligonucleotide LNA probe sequences (mir-17a) was used and hybridized at 42 °C overnight. Nonhybridized probes were washed away by wash buffer (2x SSC/0.2%SDS) for 20 min twice. Signals were detected by phosphorimager.

**CLIP analysis**. FASTQ files were trimmed using cutadapt (v1.8.3) (3′ adapter, TGGAATTCTCGGGTG CCAAG G, 5′ adapter CTACAGTCCGACGATC) to remove adapter sequences. The randomized 5′ and 3′ NN bases of the adapter were next appended to the FASTQ read ID. Reads were mapped to the human genome (hg19 build) using novoalign (v3.02.00). Reads > 25 nt after trimming were aligned with -l 25 -t 85 settings, whereas shorter reads were aligned with -l 20 -t 30 settings. PCR duplicates were removed using UMI-Tools (v. 0.5.3) with the "directional" option and alignments with MAPQ < 10 were discarded. Peaks were called by merging all overlapping alignments using bedtools merge (v2.26.0). Peaks were annotated based on overlaps with annotations downloaded from the UCSC table browser (hg19). Peaks were intersected with the following annotations in order requiring 1 bp overlap, miRNA, tRNA, snoRNA, lincRNA, utr3, utr5, cds, ncRNA, repeatMasker, introns, mitochondria, retroelements, pseudogenes, or intergenic. Peaks were classified based on the first intersecting annotation. Repeat associated regions/RNA annotations are peaks that intersect with repeatMasker annotations from UCSC genome browser. Peaks were filtered to only keep peaks present in three of five libraries, and with a minimum read count of 5. Libraries were also independently mapped to a database of pre-microRNA sequences (mirbase release 19) using blastn (v.2.2.28) to assess alignments with hairpin sequences. Publically available CLIP datasets were downloaded from GEO or ENCODE using the following accessions: DICER1 (GSE55333), DROSHA and DGCR8 (GSE61979), and DGCR8 (ENCFF491LUG).

**Small RNA analysis**. FASTQ files from DGCR8$^{cko}$ and DICER1$^{cko}$ experiments were trimmed to remove 3′ adapter sequence (TCGTATGCCGTCTTCTGCTTG)[19]. FASTQ files from XPO5$^{cko}$ experiments were trimmed to remove 3′ adapter sequence (TGGAATTCTCGG GTGCCAAGG), 5′ adapter (CTA-CAGTCCGACGATC), and randomized NN nucleotides introduced by the adapters from both the 5′ and 3′ end. Following trimming reads were then aligned with Bowtie2 (v.2.1.0) to the mouse genome (mm10) using local alignment. A custom annotation file was built containing GENCODE annotations, v19 for human, M8 for mouse with Mt_rRNA, lincRNA, misc_RNA, rRNA, sRNA, sm_RNA, snRNA, snoRNA, vaultRNA, and tRNA biotypes, and miRBase microRNA annotations (release 19). Reads overlapping annotations were counted using Htseq-count (v.0.6.0), requiring a minimum MAPQ of 10. FASTQ files from HEK293T cells were trimmed to remove 3′ adapter sequence (TGGAATTCTCGGGTGCCAAGG), 5′ adapter (CTACAGTCCGAC ATC), and randomized NN nucleotides introduced by the adapters from both the 5′ and 3′ end and aligned using Bowtie2 as described above. For comparing XPO5 HITS-CLIP to small RNA-Seq, small RNA-Seq reads were aligned with instead with novoalign (-l 18 -h 100 -t 60 -n 99) and reads were enumerated by counting intersections with a pre-miRNA database from miRbase.

**RNA secondary structure analysis**. RNA folding predictions for Fig. 5j were generated using RNAFold web server (v2.3.3). Peaks > 200 nt in length were excluded from the folding analysis. miRNA metagene plots were generated using the HTSeq python library (v0.6.0). The secondary structure of hTR in Fig. 5h was adapted from the telomerase database[34,45]. The secondary structures of noncoding RNAs in Figs. S2, S4, and S5 were generated using RNAFold web server (http://rna.tbi.univie.ac.at/cgi-bin/RNAWebSuite/RNAfold.cgi).

**Statistical information**. For all experiments with error bars, the standard deviation was calculated to indicate the variation within each experiment. Numbers of animals used for phenotype study has indicated in figures. Student's t test was used for most experiments. A two-sided Mann–Whitney test was used for Fig. 5j.

**Reporting summary**. Further information on research design is available in the Nature Research Reporting Summary linked to this article.

## Data availability

XPO5 HITS-CLIP data and small RNA-seq data are deposited in the Gene Expression Omnibus data repository with accession number GSE111964. The source data underlying Figs. 1a, 2a–e, g, i, 3a–c, 4b, d, f, g, 5c, f, 6f–h, 7d–g, l, and 8e–h and Supplementary Figs. 2b, 3, 5b and 5c are provided as a Source Data file. All data that support the findings of this study are available from the corresponding author upon reasonable request.

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

## Acknowledgements

We thank all members of the Yi laboratory for suggestions and helpful discussions, N. Pace for generous support. We thank I. Macara for XPO5 and RanQ69L expression plasmids (Vanderbilt University School of Medicine), V. N. Kim for DROSHA and DGCR8 expression plasmids (Seoul National University), B. Gao and K. Diener for sequencing. Research reported in this publication was supported by the National Institute of Arthritis and Musculoskeletal and Skin Diseases of the National Institutes of Health under Award Number R01AR059697 and R01AR066703 (to R.Y.). The content is solely the responsibility of the authors and does not necessarily represent the official views of the National Institutes of Health. J.E.L. was supported by an American Cancer Society postdoctoral fellowship (#125209). Y.Y. was supported by an NYSTEM postdoctoral fellowship. Work in E.C.L.'s group was supported by the National Institutes of Health (R01-GM083300 and R01-HL135564) and MSK Core Grant P30-CA008748.

## Author contributions

R.Y. conceived the study. J.W., J.E.L., Y.Y., E.C.L., and R.Y. designed the experiments. J.W. performed embryonic and skin studies with assistance from J.E.L. and XPO5 binding assays and pri-miRNA processing assays with assistance from S.M.M. J.E.L. established mouse models, performed initial molecular, and phenotypical studies, performed XPO5 HITS-CLIP experiments. K.R. performed bioinformatics analysis related to HITS-CLIP and miR-seq. R.Y. supervised the study, analyzed the data, and wrote the manuscript with input from all authors.

## Competing interests

The authors declare no competing interests.
