## [Peer Review File · Nature Communications]

Reviewers' comments:

Reviewer #1 (Remarks to the Author):

In the manuscript entitled “XPO5 promotes primary miRNA processing independently of RanGTP,” Wang et al. performed CLIP-seq to identify XPO5 binding sites in HEK cells. The authors demonstrated unexpected interactions between XPO5 and highly structured and clustered pri-miRNAs, both in vivo and in vitro. This interaction is independent of RanGTP, and appears to boost Microprocessor processing efficiency on these pri-miRNAs. The authors went on to characterize XPO5 KO and skin cKO mice. Developmental and skin morphogenesis defects observed in these mouse models could be attributed to the loss of miRNAs. Overall, these findings are novel and the experiments are well controlled. The manuscript could be strengthened further if the following points are addressed:

Major points:

1. A major conclusion “XPO5 facilitates the microprocessor cleavage of clustered pri-miRNAs” of this manuscript was drawn from Figure 3. However, because Flag-XPO5 was immunoprecipitated from transfected HEK293T cells, other co-precipitants, not necessarily microprocessor itself, may contribute to the elevated pri-miRNA processing efficiency. To exclude this possibility, a negative control in this set of experiments is necessary. For example, the author could mutate the hypothesized binding sites of XPO5 on the pri-miR-19a and pri-miR-15b-16-2, or use another pri-miRNA that does not bind to XPO5. The addition of Flag-XPO5 should not affect processing of these substrates. Alternatively, recombinant XPO5 purified from *E. coli* (Fig. 2) should be used.
2. Some key information should be better summarized for the readers to grasp. For example, it is not clear whether pri-miR-17~92 and pri-miR-15b-16-2 are the only pri-miRNA clusters that can bind XPO5, according to the XPO5-CLIP data. Are miR-320 and miR-484 the only miRNAs not affected by XPO5 KO in the mouse models? Is there any change for other XPO5 target RNAs in XPO5 KO mice? A summary table like Table S1 to address these questions would be very helpful.

Minor points:

1. It was mentioned in the result that “In the absence of UV crosslink or XPO5 antibody, we did not recover any RNA fragments upon immunoprecipitation.” But Fig.1a does not have results without UV crosslink.
2. The legend “small RNA seq” in Fig.1c is partially covered by the figure below.
3. The range of read density in Fig.1f should start at 0. In the current version, only Dicer Par-CLIP result is different from others.
4. It would be helpful to explain in the figure legend that the negative IGV track of Fig.1i means the reads are mapped to the minus strand. Also, the figure legend says “Data range is shown on the right of each track,” but it is on the left of each track.
5. In Fig.1j, it is interesting that within the pri-miR-17~92 cluster, pre-miR-20a has the most abundant sequencing reads in other CLIP-seq, but it has almost no reads in XPO5-CLIP. Could the authors offer possible explanations?
6. In Fig.2e, the size of the negative control symbol “-“ is different from all the other negative control

symbols.

7. The unit of Kd (nM) was missing in Fig.2f and Fig.4.

8. In Fig.3, the quantity numbers for pri-miRNA are in the middle of the gel. It would be better to have both pri-miRNA and pre-miRNA numbers at the bottom of the gel.

9. Please elaborate the statement that “the folding energy of XPO5-associated RNAs was similar to that of miRNA hairpins” (pages 13-14), while Fig.4i showed significant difference ($p=5.04e-120$) between XPO5 peaks and mirbase pre-miRNA.

10. It was stated that “neither XPO5KI nor XPOKO breedings produced any XPO5 null mice at birth, ” but XPOKO was not shown in Fig.5b.

11. In “Cell culture” Method, “CO₂” should be “CO₂.”

12. In “RNA Purification and mRNA and miRNA qPCR” method part, it was stated that “Differences between samples and controls were calculated using the method. ” But it was in Fig.5f, 5h and Fig.7f.

13. In HITS-CLIP method description, it seems odd to label the 3' end of the RNA with 32P-ATP using PNK. Please validate that it is not a typo.

Reviewer #2 (Remarks to the Author):

Wang et al.

XPO5 promotes primary miRNA processing independently of RanGTP

In this manuscript, the authors have profiled the RNA substrate spectrum of XPO5 using HITS-CLIP. As expected, most pre-miRNAs can be identified. Interestingly, XPO5 appears to bind not only the hairpin of pre-miRNAs but also primary miRNA transcripts outside of the hairpin region. This is most evident on the miR-17-92 primary transcript. The authors performed in vitro binding experiments and report that binding to such unexpected regions is independent of RanGTP, which is required for canonical pre-miRNA export. Furthermore, XPO5 stimulates microprocessor cleavage of pri-miRNAs in vitro. XPO5 is not only crosslinked to miRNA transcripts but also to many other structured RNAs including vault RNAs or scaRNAs. The authors find that double stranded regions within these RNAs serve as binding platforms for XPO5. To further characterize the role of XPO5 in vivo, Wang et al. generated knock in as well as conditional knock out mice. They find that the knock out is embryonic lethal and this is due to strongly reduced production of miRNAs and thus differentiation deficits. A skin-specific knock out revealed that XPO5 is important for skin morphogenesis and is required for mature miRNA biogenesis in this tissue.

This is a well-written and well-presented study on the function of XPO5. Although the role of XPO5 in miRNA export is known for a long time and well established, tissue culture knock-out models showed that many miRNAs are still expressed in the absence of XPO5 albeit at lower levels. Thus, the presented study in animals helps solving this issue. In addition, the manuscript suggests additional, potentially miRNA-independent functions of XPO5. However, the function of XPO5 on other RNA species remains elusive and is not very well established in this manuscript. More detailed points and issues are listed

below.

1. Figure 1A. The authors mention in the text ‘...the absence of UV crosslink...’. Where is this shown in the Figure?

2. The biochemical experiments appear very solid and convincing. However, there are a number of additional controls that should be added. For example, in Figure 2B, how would a control RNA of this length (maybe containing some random secondary structures and/or local hairpins) look like in these in vitro binding assays? This would be an important specificity control.

3. Also the stimulation of microprocessor cleavage looks very convincing. For clarity, why did the authors IP tagged XPO5 from cell lines instead of using the recombinant protein used for the EMSA experiments? This should be mentioned because recombinant protein might be cleaner (although there is no activity of the IP XPO5).

4. Would XPO5 also stimulate processing of non-miRNA substrates such as vault RNA or other non-coding RNAs? It is not fully clear whether XPO5 binding leads to the production of Ago-associated small RNAs. As stated correctly, such RNAs are often found in RNAseq experiments. However, they are often of rather low abundance and their functional relevance remains unclear.

5. I think the main weakness of this manuscript is that most of the non-canonical binding sites are not functionally validated and thus it remains elusive whether or not they are functionally relevant. Does binding of XPO5 to miRNA-independent substrates contribute to the phenotypes that are observed in mice? The manuscript in its present form does not convincingly show that these non-canonical targets are really functionally relevant or only unspecific background binding. Maybe the authors could add more functional validation data to this part.

Reviewer #3 (Remarks to the Author):

In this manuscript Wang et al aimed to characterize the miRNA-biogenesis-dependent and independent functions of XPO5 during early embryonic development.

The presented data are technically sound and demonstrate that:

1. XPO5 binds - in addition to its previously extensively characterized binding to pre-miRNAs – binds to many cellular RNAs with double-stranded regions suggesting possible functions for XPO5 that are independent of miRNA-biogenesis.
2. XPO5 binding to miRNA precursors, in particular to members of the miR-17-92 and miR-15/16 cluster, enhances the processing efficiency
3. The binding of XPO5 to pri-miR19a and pri-miR-15-16 is independent of RanGTP
4. Genetic deletion of XPO5 results in embryonic lethal phenotype. XPO5 KO embryos are smaller at E6.5 and fail to initiate organogenesis by E8.5.

5. Conditional ablation of XPO5 in the skin results in impaired skin development and neonatal lethality between postnatal days 5 and 18.

6. The deletion of XPO5 resulted in impaired miRNA biogenesis, impaired hair follicle formation and growth in the skin, similar to the genetic deletion of other genes, which are also part of the miRNA-processing machinery.

In addition, the results suggest that XPO5 preferentially binds to primary transcripts of polycistronic miRNAs over transcripts of monocistronic miRNAs, although it is not clear whether this conclusion is generalizable.

Comments:

1. One of the most significant new findings of this study is the apparently pervasive binding of XPO5 to cellular transcripts containing dsRNA regions. It is not clear however how much of this is due to a specific binding, i.e. has some kind of specific biological role and how much is an unspecific ability of the protein due to its affinity to RNAs and association with RNA-binding proteins. Can the authors exclude the possibility that XPO5-binding to many of the dsRNAs is "off-target" effect? Does XPO5-binding profile change during embryonic development of skin differentiation?

2. The similarity of the phenotype of XPO5 KO and cKO mice to the phenotypes of Dicer, Dgcr8 and Ago deletions weakens the conclusions about the possible miRNA-biogenesis-independent functions of XPO5. One could ask: Why is the phenotype of XPO5-KO mice so similar to the phenotypes resulting from the deletion of genes regulating miRNA-biogenesis? Does this not imply that the most important biological function of XPO5 is still related to the miRNA biogenesis/export? Did the full body XPO5 KO or the XPO5 cKO mice have any phenotype that could not be explained by the effect on the miRNA-biogenesis? This is an important issue, because the novelty of the findings according to the authors is the previously unrecognized role for XPO5 in addition to the nuclear export of miRNA precursors.

3. The authors conclude that XPO5 binds preferentially to multicistronic miRNA precursors in which miRNAs are closely spaced over monocistronic miRNA precursors. I am not sure whether this conclusion is generalizable. More miRNA precursors representing both classes should be examined to draw a generalized conclusion. It is not clear from the data at present whether this is only due to the fact that longer RNAs may contain more dsRNA region than shorter ones.

4. Can the authors identify any sequence motif associated with XPO5-binding in addition to that it binds to dsRNAs?

5. Figure 2b: why does the increasing amount of XPO5 change the molecular weight of the complex rather than only altering the intensity of the signal?

6. Figure 2e: please double check this image, the signal seems to be clearly different in the lower half of the image. Is this an artefact?

7. What is the functional consequence of XPO5-binding to non-miRNA targets? Does XPO5 KD alter the subcellular localization of its non-canonical target transcripts? Is it involved in the processing of the non-canonical targets?

8. Does XPO5 bind to mRNAs? If yes, have the authors analysed changes in transcript and protein level of XPO5-bound transcripts in full-body and skin-specific XPO5 KO mice?

9. Do HITS-CLIP signals correlate with the abundance of the transcripts? What strategy has been utilized to exclude the possibility that the pervasive binding of XPO5 to transcripts is not unspecific? Please

explain.

10. Does XPO5 associate with Drosha and/or Dgcr8 in cells?

11. Figure 6: The effect of XPO5-deletion in the skin should be addressed in more detail. qPCR results should be added to support conclusions with K5 and K1 antibodies only.

12. Were there any differences in the granular layer in the epidermis in XPO5-deficient mice (Figure 6)?

13. Did the authors analysed changes in other K14-expressing tissues of the mice?

14. The authors show that the two skin miRNAs, miR-203 and miR-205 display altered level in the skin of XPO cKO mice. What about the level of the member of the miR-17-92 cluster and miR-15/16 cluster miRNAs? The authors should study also the level of pre/pri miRNAs not only the mature miRNAs to find out whether the processing was affected?

15. The authors should validate changes in the non-canonical XPO5-target genes in full-body and skin-specific XPO KO mice.

Reviewers' comments:

Reviewer #1 (Remarks to the Author):

In the manuscript entitled "XPO5 promotes primary miRNA processing independently of RanGTP," Wang et al. performed CLIP-seq to identify XPO5 binding sites in HEK cells. The authors demonstrated unexpected interactions between XPO5 and highly structured and clustered pri-miRNAs, both in vivo and in vitro. This interaction is independent of RanGTP, and appears to boost Microprocessor processing efficiency on these pri-miRNAs. The authors went on to characterize XPO5 KO and skin cKO mice. Developmental and skin morphogenesis defects observed in these mouse models could be attributed to the loss of miRNAs. Overall, these findings are novel and the experiments are well controlled. The manuscript could be strengthened further if the following points are addressed:

We thank the reviewer's positive evaluation and constructive criticism, and we have addressed his/her concerns below.

Major points:

1. A major conclusion "XPO5 facilitates the microprocessor cleavage of clustered pri-miRNAs" of this manuscript was drawn from Figure 3. However, because Flag-XPO5 was immunoprecipitated from transfected HEK293T cells, other co-precipitants, not necessarily microprocessor itself, may contribute to the elevated pri-miRNA processing efficiency. To exclude this possibility, a negative control in this set of experiments is necessary. For example, the author could mutate the hypothesized binding sites of XPO5 on the pri-miR-19a and pri-miR-15b-16-2, or use another pri-miRNA that does not bind to XPO5. The addition of Flag-XPO5 should not affect processing of these substrates. Alternatively, recombinant XPO5 purified from *E. coli* (Fig. 2) should be used.

To confirm the specificity of our experiments, we first generated two pri-mir-19a mutations, which delete one (pri-mir-19a truncation v1) or both (pri-mir-19a truncation v2) basal stem regions. We found that pri-mir-19a truncation v1 can be bound by one XPO5 molecule in a RanGTP-independent manner, demonstrated by a single-shift band in the binding assay (new Fig. 2h); and pri-mir-19a truncation v2 is no longer bound by XPO5 (new Fig. 2j). Importantly, although both mutants can be cleaved by the DRISHA/DGCR8 microprocessor, only pri-mir-19a truncation v1, which can be bound by XPO5, is more efficiently cleaved in the presence of XPO5 (new Fig. 3b-c). We have further generated pri-mir-15b from the pri-mir-15b-16-2 cluster. We found that pri-mir-15b, which also contains one basal stem region, is bound by one XPO5 in a RanGTP-independent manner (new Fig. 4c-e). And DRISHA/DGCR8-mediated cleavage of pri-mir-15b is also enhanced by XPO5 (new Fig. 4g). These data not only confirm the specificity of XPO5 for promoting the nuclear cleavage but also pinpoint the basal stem regions as XPO5 binding sites on pri-miRNA.

2. Some key information should be better summarized for the readers to grasp. For example, it is not clear whether pri-miR-17~92 and pri-miR-15b-16-2 are the only pri-miRNA clusters that can bind XPO5, according to the XPO5-CLIP data. Are miR-320 and miR-484 the only miRNAs not affected by XPO5 KO in the mouse models? Is there any change for other XPO5 target RNAs in XPO5 KO mice? A summary table like Table S1 to address these questions would be very helpful.

In revision, we generate a new Table S2 for the XPO5-CLIP data, listing all mapped RNA species and their genome coordinates as well as reads abundance.

Due to the difficult to quantify miRNA depletion with smRNA-seq, we only focus on most abundantly expressed miRNAs in the skin, similar to what we have done for Dgcr8 and Dicer1 KO. Among these miRNAs, miR-320 and miR-484 are the only ones that are not strongly affected by XPO5 KO, as we have observed for Dgcr8 KO in the skin previously (Yi et al., PNAS 2009).

For other non-miRNA species, we have tested vtRNA and did not observe any changes (new Fig. 5f). More comprehensive analysis to these new binding targets of XPO5 will be comprehensively examined in our future studies.

Minor points:

1. It was mentioned in the result that “In the absence of UV crosslink or XPO5 antibody, we did not recover any RNA fragments upon immunoprecipitation.” But Fig.1a does not have results without UV crosslink.

We apologize for the mistake in the text. In the experiments to generate XPO5-CLIP data, we only used IgG as a control. The aforementioned no UV crosslink experiments were performed at the early stage of this project. We only had the record of the experiment failed to isolate any RNA but could not locate the gel image. In revision, we change the text to “In the absence of XPO5 antibody....” to accurately describe the experiment.

2. The legend “small RNA seq” in Fig.1c is partially covered by the figure below.

We revised the panel in Fig. 1c.

3. The range of read density in Fig.1f should start at 0. In the current version, only Dicer Par-CLIP result is different from others.

The range of read density in Fig. 1f has been revised to start at 0.

4. It would be helpful to explain in the figure legend that the negative IGV track of Fig.1i means the reads are mapped to the minus strand. Also, the figure legend says “Data range is shown on the right of each track,” but it is on the left of each track.

We now noted the negative value means the reads are mapped to the minus strand, and we corrected the error in annotation in the figure legend of Fig. 1i.

5. In Fig.1j, it is interesting that within the pri-miR-17~92 cluster, pre-miR-20a has the most abundant sequencing reads in other CLIP-seq, but it has almost no reads in XPO5-CLIP. Could the authors offer possible explanations?

Based on our study, XPO5 binds to pre-miRNA hairpin in a RanGTP-dependent manner, mediating the nuclear export of these precursors, whereas XPO5 binds to some pri-miRNA in a RanGT-independent manner, promoting the microprocessor cleavage. Our XPO5-CLIP

data provide a snapshot for the overall picture of XPO5-associated RNAs. Because XPO5 has extensive association with the pri-mirna regions of the mir17~92 cluster, its association with pre-mirna precursor is obscured. In fact, there are an abundant of reads that are on pre-mir-20a, but the flanking regions simply have many more reads. This pattern is also observed in pri-mir-15a-16-1 and pri-mir-15b-16-2 regions (Fig. S1d). We think that it reflects the affinity of XPO5 to the pri-mirna regions. We added a discussion for this observation in the text (pg. 8, line 8).

6. In Fig.2e, the size of the negative control symbol “-” is different from all the other negative control symbols.

The symbol “-” has been changed to make it consistence with other negative control symbols in all panels.

7. The unit of Kd (nM) was missing in Fig.2f and Fig.4.

We have added the unit of Kd in all panels that calculate the Kd including Fig.2f, 2i, 4e and 5d.

8. In Fig.3, the quantity numbers for pri-miRNA are in the middle of the gel. It would be better to have both pri-miRNA and pre-miRNA numbers at the bottom of the gel.

This Fig. 3 now becomes Fig. 4f. Because we would like to quantify both the singly cropped pri-mir-15b-16-2 and the uncropped pri-mir as well as the pre-mir, we decided to keep the quantification in the middle of the gel for clarity. For all other processing assays (Fig. 3a-c and Fig. 4g), we moved the quantification number to the top and the bottom of the gel.

9. Please elaborate the statement that “the folding energy of XPO5-associated RNAs was similar to that of miRNA hairpins” (pages 13-14), while Fig.4i showed significant difference ($p=5.04e-120$) between XPO5 peaks and mirbase pre-miRNA.

Indeed, there is a difference between the folding energy of XPO5 bound regions and that of pre-miRNA hairpins. To clarify this point, we now separated pre-miR regions bound by XPO5 from other regions bound by XPO5 (repeat regions and non-repeat regions) and calculated their folding energy. As shown in the new Fig. 5j, pre-miRNA regions bound by XPO5 have similar folding energy to that of pre-miRNA from mirbase whereas other regions bound by XPO5 has an even lower folding energy than both. This indicates that XPO5 bound regions are more likely to have a lower folding energy than pre-miRNA hairpins. We also revised the text to reflect these observations.

10. It was stated that “neither XPO5KI nor XPOKO breedings produced any XPO5 null mice at birth, ” but XPOKO was not shown in Fig.5b.

We have carefully checked our record, the shown data – 33 WT, 80 het, 0 KO – were obtained from 20 litters of KO/+ (het) breeding. We have also compiled our KI/+ breeding

data and they showed the similar pattern as KO. The new data is now included in new Fig. 6b.

11. In “Cell culture” Method, “CO2” should be “CO₂.”

We have changed CO2 to CO₂ in the Method part.

12. In “RNA Purification and mRNA and miRNA qPCR” method part, it was stated that “Differences between samples and controls were calculated using the method. ” But it was in Fig.5f, 5h and Fig.7f.

We have deleted the repetitive state in the method part.

13. In HITS-CLIP method description, it seems odd to label the 3' end of the RNA with 32P-ATP using PNK. Please validate that it is not a typo.

We have corrected it to “labelling the 5' end of the RNA with 32P-ATP using PNK”.

Reviewer #2 (Remarks to the Author):

Wang et al.

XOP5 promotes primary miRNA processing independently of RanGTP

In this manuscript, the authors have profiled the RNA substrate spectrum of XPO5 using HITS-CLIP. As expected, most pre-miRNAs can be identified. Interestingly, XPO5 appears to bind not only the hairpin of pre-miRNAs but also primary miRNA transcripts outside of the hairpin region. This is most evident on the miR-17-92 primary transcript. The authors performed in vitro binding experiments and report that binding to such unexpected regions is independent of RanGTP, which is required for canonical pre-miRNA export. Furthermore, XPO5 stimulates microprocessor cleavage of pri-miRNAs in vitro. XPO5 is not only crosslinked to miRNA transcripts but also to many other structured RNAs including vault RNAs or scaRNAs. The authors find that double stranded regions within these RNAs serve as binding platforms for XPO5. To further characterize the role of XPO5 in vivo, Wang et al. generated knock in as well as conditional knock out mice. They find that the knock out is embryonic lethal and this is due to strongly reduced production of miRNAs and thus differentiation deficits. A skin-specific knock out revealed that XOP5 is important for skin morphogenesis and is required for mature miRNA biogenesis in this tissue.

This is a well-written and well-presented study on the function of XPO5. Although the role of XPO5 in miRNA export is known for a long time and well established, tissue culture knock-out models showed that many miRNAs are still expressed in the absence of XPO5 albeit at lower levels. Thus, the presented study in animals helps solving this issue. In addition, the manuscript suggests additional, potentially miRNA-independent functions of XPO5. However, the function of XOP5 on other RNA species remains elusive and is not very well established in this manuscript. More detailed points and issues are listed below.

We thank the reviewer's positive evaluation and constructive criticism, and we have addressed his/her concerns below.

1. Figure 1A. The authors mention in the text '...the absence of UV crosslink...'. Where is this shown in the Figure?

We apologize for the mistake in the text. In the experiments to generate XPO5-CLIP data, we only used IgG as a control. The aforementioned no UV crosslink experiments were performed at the early stage of this project. We only had the record of the experiment failed to isolate any RNA but could not locate the gel image. In revision, we change the text to "In the absence of XPO5 antibody...." to accurately describe the experiment.

2. The biochemical experiments appear very solid and convincing. However, there are a number of additional controls that should be added. For example, in Figure 2B, how would a control RNA of this length (maybe containing some random secondary structures and/or local hairpins) look like in these in vitro binding assays? This would be an important specificity control.

We appreciate the reviewer's concern for a negative control for the binding and subsequent function of XPO5. Instead of generating an artificial negative control which does not provide additional information for the role of XPO5 in RNA binding and miRNA biogenesis, we generated two pri-mir-19a truncated mutations, gradually trimming away the basal stem regions, and tested their binding by XPO5 and their cleavage by the microprocessor in the presence and absence of XPO5. We found that pri-mir-19a truncation v1 can be bound by one XPO5 molecule, demonstrated by a single-shift band in the binding assay (new Fig. 2h); and pri-mir-19a truncation v2 is no longer bound by XPO5 in a RanGTP-independent manner (new Fig. 2j). Importantly, although both mutants can be cleaved by the DROSHA/DGCR8 microprocessor, only pri-mir-19a truncation v1, which can be bound by XPO5, is more efficiently cleaved in the presence of XPO5 (new Fig. 3b-c). We also generated a new pri-mir-15b RNA containing a double-stranded basal stem region (new Fig. 4c). This pri-miRNA is again bound by XPO5 in the absence of RanGTP (new Fig. 4d-e), and its microprocessor cleavage is also enhanced (new Fig. 4g). These data not only confirm the binding specificity of XPO5 but also pinpoint the basal stem regions as XPO5 binding sites on multiple pri-miRNAs.

3. Also the stimulation of microprocessor cleavage looks very convincing. For clarity, why did the authors IP tagged XPO5 from cell lines instead of using the recombinant protein used for the EMSA experiments? This should be mentioned because recombinant protein might be cleaner (although there is no activity of the IP XPO5).

The binding buffer and the processing buffer is different in that the processing buffer contains 64mM Mg²⁺ whereas the binding buffer contains 2mM Mg²⁺. We have tested recombinant XPO5 for the processing assay. Although we still

observed more efficient reduction of unprocessed pri-mir-19a, recombinant XPO5 seems to trigger RNA degradation (see the attached image). We suspect that some contaminants in

proteins purified from bacteria contribute to this. Because our additional pri-mir-19a mutations and pri-mir-15b have confirmed the specificity of our IP tagged XPO5, we choose to present the comprehensive data generated from IP tagged XPO5 for the processing experiments.

4. Would XPO5 also stimulate processing of non-miRNA substrates such as vault RNA or other non-coding RNAs? It is not fully clear whether XPO5 binding leads to the production of Ago-associated small RNAs. As stated correctly, such RNAs are often found in RNAseq experiments. However, they are often of rather low abundance and their functional relevance remains unclear.

We share the reviewer's interests to study non-miRNA substrates. Considering our focus on studying the role of XPO5 in miRNA processing independent of nuclear export and demonstrating the requirement of XPO5 for mammalian development, we believe these additional studies should be carried out comprehensively in future investigation. That being said, we have generated new XPO5 KO MEF cells and probed the production and nuclear/cytoplasmic localization of vtRNA. We observed that neither processing nor nuclear/cytoplasmic localization of vtRNA is altered upon the deletion of XPO5. These data suggest that RanGTP-independent binding of vtRNA by XPO5 does not affect the biogenesis, localization and stability of vtRNA. In the revised manuscript, we show these data in Fig. 5f. We will study the functional consequence of XPO5 binding to vtRNA and other non-miRNA substrates in future studies.

5. I think the main weakness of this manuscript is that most of the non-canonical binding sites are not functionally validated and thus it remains elusive whether or not they are functionally relevant. Does binding of XPO5 to miRNA-independent substrates contribute to the phenotypes that are observed in mice? The manuscript in its present form does not convincingly show that these non-canonical targets are really functionally relevant or only unspecific background binding. Maybe the authors could add more functional validation data to this part.

We share the reviewer's interests to investigate non-miRNA substrates. As we showed in response to the reviewer's previous point, we tested the effect of XPO5 loss on vtRNA but did not observe any changes. However, we'd like to point out that our main focus and novel finding in this study is to study the nuclear export-independent function of XPO5 during miRNA biogenesis and provide genetic and molecular evidence for the requirement of XPO5 for mammalian development. Through our HITS-CLIP study of XPO5 and extensive biochemical analysis, we have revealed a new function of XPO5 in binding to pri-miRNA, specifically in the basal stem region, and promote DROSHA/DGCR8-mediated cleavage of pri-mir-19a and pri-mir-15b-16-2. We believe these conclusions are important and well supported by our experimental evidence, and we hope the reviewer agrees with us that these new insights warrant the publication of our manuscript at Nature Communications. We have revised the manuscript to point out that it remains unclear whether XPO5 plays a role in non-miRNA substrates and we wrote carefully that the defects observed in XPO5 KO may not only reflect the defective miRNA pathway (pg. 19, the end of result section).

Reviewer #3 (Remarks to the Author):

In this manuscript Wang et al aimed to characterize the miRNA-biogenesis-dependent and independent functions of XPO5 during early embryonic development.

The presented data are technically sound and demonstrate that:

1. XPO5 binds - in addition to its previously extensively characterized binding to pre-miRNAs – binds to many cellular RNAs with double-stranded regions suggesting possible functions for XPO5 that are independent of miRNA-biogenesis.
2. XPO5 binding to miRNA precursors, in particular to members of the miR-17-92 and miR-15/16 cluster, enhances the processing efficiency
3. The binding of XPO5 to pri-miR19a and pri-miR-15-16 is independent of RanGTP
4. Genetic deletion of XPO5 results in embryonic lethal phenotype. XPO5 KO embryos are smaller at E6.5 and fail to initiate organogenesis by E8.5.
5. Conditional ablation of XPO5 in the skin results in impaired skin development and neonatal lethality between postnatal days 5 and 18.
6. The deletion of XPO5 resulted in impaired miRNA biogenesis, impaired hair follicle formation and growth in the skin, similar to the genetic deletion of other genes, which are also part of the miRNA-processing machinery.

In addition, the results suggest that XPO5 preferentially binds to primary transcripts of polycistronic miRNAs over transcripts of monocistronic miRNAs, although it is not clear whether this conclusion is generalizable.

Comments:

1. One of the most significant new findings of this study is the apparently pervasive binding of XPO5 to cellular transcripts containing dsRNA regions. It is not clear however how much of this is due to specific binding, i.e. has some kind of specific biological role and how much is an unspecific ability of the protein due to its affinity to RNAs and association with RNA-binding proteins. Can the authors exclude the possibility that XPO5-binding to many of the dsRNAs is “off-target” effect? Does XPO5-binding profile change during embryonic development of skin differentiation?

To test the specificity of XPO5 binding and further narrow down the bound region, we performed a series of binding assays by generating two pri-mir-19a mutations. These two mutations delete one (pri-mir-19a truncation v1) or both (pri-mir-19a truncation v2) basal stem regions, respectively. We found that pri-mir-19a truncation v1 can be bound by one XPO5 molecule in a RanGTP-independent manner, demonstrated by a single-shift band in the binding assay (new Fig. 2h); and pri-mir-19a truncation v2 is no longer bound by XPO5 (new Fig. 2j). Importantly, although both mutants can be cleaved by the DRISHA/DGCR8 microprocessor, only pri-mir-19a truncation v1, which can be bound by XPO5, is more efficiently cleaved in the presence of XPO5 (new Fig. 3b-c). We have further generated pri-mir-15b from the pri-mir-15b-16-2 cluster. We found that pri-mir-15b, which also contains one basal stem region, is bound by one XPO5 in a RanGTP-independent manner (new Fig. 4c-e). And DRISHA/DGCR8-mediated cleavage of pri-mir-15b is also enhanced by XPO5 (new Fig. 4g). These data not only confirm the specificity of XPO5 for promoting the nuclear cleavage but also pinpoint the basal stem regions as XPO5 binding sites on pre-miRNA.

We also further separated XPO5 bound RNA into repeat regions, non-repeat regions and pre-mirna regions and compared their folding energy. As shown in new Fig. 5j, non-pre-

miRNA regions that are bound by XPO5 have even lower folding energy than pre-miRNA hairpin. These data argue that XPO5 indeed has a preference to dsRNA regions.

Currently, XPO5 HITS-CLIP does not work with very limited samples from mouse embryonic development, therefore we can't test whether XPO5-binding profile changes during embryonic skin development.

2. The similarity of the phenotype of XPO5 KO and cKO mice to the phenotypes of Dicer, Dgcr8 and Ago-deletions weaken the conclusions about the possible miRNA-biogenesis-independent functions of XPO5. One could ask: Why is the phenotype of XPO5-KO mice so similar to the phenotypes resulted by the deletion of genes regulation miRNA-biogenesis? Does this not imply that the most important biological function of XPO5 is still related to the miRNA biogenesis/export? Did the full body XPO5 KO or the XPO cKO mice have any phenotype that could not be explained by the effect on the miRNA-biogenesis? This is an important issue, because the novelty of the findings is according to the authors is the previously unrecognized role for XPO5 in addition to the nuclear export of miRNA precursors.

We share the reviewer's interests to study non-miRNA substrates. Considering our focus on studying the role of XPO5 in miRNA processing independent of nuclear export and demonstrating the requirement of XPO5 for mammalian development, we believe these additional studies should be carried out comprehensively in future investigation. Both the title and the abstract of our manuscript focus on miRNA-related but new functions of XPO5. The novelty of our study is centered on the identification of XPO5 binding to several clustered pri-mirna, our comprehensive study of the binding of pri-mirna precursors by XPO5 in a RanGTP-independent manner, and such binding by XPO5 promotes DROSHA/DGCR8-mediated cleavage.

That being said, we have generated new XPO5 KO MEF cells and probed the production and nuclear/cytoplasmic localization of vtRNA. We observed that neither processing nor nuclear/cytoplasmic localization of vtRNA is altered upon the deletion of XPO5. These data suggest that RanGTP-independent binding of vtRNA by XPO5 does not affect the biogenesis, localization and stability of vtRNA. In the revised manuscript, we show these data in Fig. 5f. We will study the functional consequence of XPO5 binding to vtRNA and other non-miRNA substrates in future studies.

In addition, we revised the manuscript to discuss the possibility that the phenotype of XPO5 may not be caused entirely by the loss of miRNAs (pg. 19, the end of result section).

3. The authors conclude that XPO5 binds preferentially to multicistronic miRNA precursors in which miRNAs are closely spaced over monocistronic miRNA precursors. I am not sure whether this conclusion is generalizable. More miRNA precursors representing both classes should be examined to draw a generalized conclusion. It is not clear from the data at present whether this is only due to the fact that longer RNAs may contain more dsRNA region than shorter ones.

We do not intend to suggest that the binding of XPO5 to pri-mir-17~92, pri-mir-15a-16-2 and pri-mir-15b-16-1 can be generalized to other clustered miRNAs. For example, we already

showed that we did not observe XPO5-CLIP reads on pri-mir-200b/a-429 (Supplementary Fig. 1e). This suggests the binding of XPO5 is dependent upon local structures and different pri-mirna may have different structures. We made this note in the introduction (pg. 4, last paragraph), the result section (pg. 9, first paragraph), and discussion (pg. 20, the end of the first paragraph).

4. Can the authors identify any sequence motif associated with XPO5-binding in addition to that it binds to dsRNAs?

We have performed de novo motif analysis with our XPO5-associated HITS-CLIP reads, and we did not recover strong motifs. This observation is consistent with the idea that XPO5 does not have a preference to any motif other than a general preference to dsRNA regions, as one may expect for a general factor of miRNA biogenesis. Similarly, Dicer1 or DROSHA/DGCR8 are not reported to bind any RNA in a sequence motif-specific manner.

5. Figure 2b: why does the increasing amount of XPO5 changes the molecular weight of the complex rather than only altering the intensity of the signal?

In gel shift assay, increased molecular weight reflects the binding between two or more molecules, in our case the binding between pri-mir and XPO5. Because the signal comes from radio-labeled RNA, the signal intensity only reflects the amount/percentage of RNA that is bound by XPO5. Whereas the size reflects the amount of XPO5 that can bind to the RNA. If only a specific amount of XPO5 can bind to a single RNA as in the case of XPO5 binding to pre-miRNA in the presence of RanGTP, a single shifted band should be observed. The changed (increased) molecular weight in Fig. 2b-c indicates that many XPO5 molecules bind to pri-mir-17~92 when the concentration of XPO5 is increased. These super-shifted bands therefore indicate a single pri-mir17~92 RNA can bind to many XPO5, consistent with our HITS-CLIP dataset.

6. Figure 2e: please double check this image, the signal seems to be clearly different in the lower half of the image. Is this an artefact?

This result was indeed from an intact image but probably affected by scanning. We have repeated this experiment and generated a better image (Fig. 2e). In addition, uncropped image is also provided as a source file.

7. What is the functional consequence of XPO5-binding to non-miRNA targets? Does XPO5 KD alters the subcellular localization of its non-canonical target transcripts? Is it involved in the processing of the non-canonical targets?

To answer some of these questions, we generated XPO5 KO MEF cells and examined vtRNA. As shown in new Fig. 5f, we observed no changes in vtRNA levels or the nuclear/cytoplasmic localization. These results suggest that XPO5 does not affect the level or localization of vtRNA.

8. Does XPO5 bind to mRNAs? If yes, have the authors analysed changes in transcript and protein level of XPO5-bound transcripts in full-body and skin-specific XPO5 KO mice?

A few XPO5 binding reads are mapped to mRNA. We note, however, HITS-CLIP analysis was done in human 293T cells, and this precludes the analysis of the effect of XPO5 KO on these transcripts in mice. Our current manuscript focuses on nuclear export-independent function of XPO5 in miRNA biogenesis and the requirement of XPO5 during mouse embryonic and skin development. The analysis of XPO5's role in non-miRNA substrates including mRNAs will be carried out in future studies.

9. Do HITS-CLIP signals correlate with the abundance of the transcripts? What strategy has been utilized to exclude the possibility that the pervasive binding of XPO5 to transcripts is not unspecific? Please explain.

HITS-CLIP results are generally affected by the abundance of RNA substrates and the affinity between RNA and protein. We have examined whether XPO5 non-specifically binds to any RNA simply due to the RNA abundance. Several lines of evidence argue against this notion. First, for abundantly recovered RNA species in our XPO5 HITS-CLIP dataset, we generally identify double-stranded regions but not the entire transcript. In the manuscript, we already examined mir-31 (Fig. 1i), Dgcr8 (Fig. 1k) and hTR (Fig. 5g). In these transcripts, only pre-mir hairpin regions (mir-31 and Dgcr8) and dsRNA regions (for hTR) have XPO5 reads, serving as the best control for specificity. Second, we generated several pri-mir-19a mutants with different basal stem regions and mapped XPO5 binding sites to the double-stranded basal regions (new Fig. 2e, h, j). In addition, we have measured the Kd of XPO5 to many substrates that are between 7.99nM and 36.27nM (Fig. 2, 4, 5), these data suggest that XPO5 has a strong affinity to these representative substrates including pri-mir and vtRNA. Third, our new folding energy analysis indicate that XPO5 bound repeat and non-repeat regions have even lower folding energy than pre-miRNA hairpins (new Fig. 5j). Finally, we have downloaded a RNA-seq dataset generated from HEK293T cells (Zhang XQ and Yang JH, Methods Mol Biol 2018; GEO accession GSM2563627). The top 10 most abundantly expressed RNAs are: 7SK RNA, 7SL1 RNA, 7SL2 RNA, RPPH1, SNORD3B-1, RMRP, SNORD3A, HIST2H4B and HIST2H4A. Among them, we indeed detected XPO5 HITS-CLIP signals on 7SK (Fig. S4g), 7SL1 (Fig. 5i), 7SL2, RPPH1, RMRP and SNORD3A (not shown), but we also didn't observe any signals on SNORD3B-1, HIST2H4B and HIST2H4A. In addition, even within these small RNAs, XPO5 associated regions are predicted to form dsRNA (see Fig. S4g, 7SK RNA). Furthermore, pri-mir-17~92 cluster, which is one of the strongest XPO5-associated RNAs, ranked 3,359 among all detected RNAs. Collectively, these data suggest that our XPO5 HITS-CLIP specifically identify dsRNA regions.

10. Does XPO5 associate with Drosha and/or Dgcr8 in cells?

No. XPO5 does not associate with Drosha or Dgcr8. This is consistent with our observation that IPed XPO5 does not cleave pri-miRNA. In addition, we didn't detect XPO5 in DROSHA and DGCR8 IP samples by Western blot and silver stain (shown in the gel on the left and in new Fig. S3, pg. 11).

11. Figure 6: The effect of XPO5-deletion in the skin should be addressed in more detail. qPCR results should be added to support conclusions with K5 and K1 antibodies only.

We measured the Krt1, Krt5, Lor and beta-4 mRNA levels by qPCR in XPO5 cKO skin. The levels of those genes are not differentially changed, as shown in Fig. 7l.

12. Were there any differences in the granular layer in the epidermis in XPO5-deficient mice (Figure 6).

We haven't observed any differences in the granular layer in the epidermis in XPO5-deficient mice. This observation is similar to those observed in Dicer1 and Dgcr8 cKO skin, in which the ablation of miRNA pathway does not alter epidermal differentiation.

13. Did the authors analyzed changes in other K14-expressing tissues of the mice?

We didn't extend our study beyond the epithelial cells of the skin.

14. The authors show that the two skin miRNAs, miR-203 and miR-205 display altered level in the skin of XPO cKO mice. What about the level of the member of the miR-17-92 cluster and miR-15/16 cluster miRNAs? The authors should study also the level of pre/pri miRNAs not only the mature miRNAs to find out whether the processing was affected?

The levels of miRNAs from miR-17-92 and miR15b-16 are all down-regulated measured by qPCR, as shown in new Figure 8e-f. For the pre/pri-miRNAs of mir-17, we also performed northern blot using total RNA isolated from XPO5 cKO skin epidermal cells, as shown in new Fig. 8g. We observed strong depletion of mature mir-17, consistent with our miR-seq and qPCR results. But we did not observe strong changes in pri/pre-miRNA. This observation is similar to our previous study of Dicer1 cKO, in which we usually don't observe the accumulation of pre-miRNA despite the loss of Dicer1 (Yi et al., Nature Genetics 2006). It is possible pri/pre-mirna are under active regulation, and in the absence of XPO5 they could become unstable or degraded. Finally, within a cell many known and unknown mechanisms can affect the accumulation of pri/pre-miRNA species. We believe that cell free, biochemical assays remain the best option to study how XPO5 may affect pri-miRNA processing, as we have extensively documented in this manuscript.

15. The authors should validate changes in the non-canonical XPO5-target genes in full-body and skin-specific XPO KO mice.

To investigate non-miRNA targets of XPO5, we generated XPO5 KO MEF cells and measured vtRNA levels and localization in these cells by northern blot. We did not observe any changes in the expression level and localization. This suggests that XPO5 binding to vtRNA does not change vtRNA levels or accumulation. We believe a thorough investigation, which is beyond the scope of our current study, is warranted for future studies.

REVIEWERS' COMMENTS:

Reviewer #1 (Remarks to the Author):

The revised manuscript addressed most of the criticisms raised in the first review. I recommend this manuscript to be published with a few minor improvements:

1. It is unclear why quantification of miRNA depletion is particularly difficult. Readers should be able to evaluate miRNA level changes in XPO5 KO if the processed data to graph Fig. 8d is given.
2. In Fig.5j-related Main text or Methods section, the authors should define "repeat associated regions/RNA".

Reviewer #2 (Remarks to the Author):

The authors have addressed all points that I had raised on the previous version of the manuscript. They have clarified parts of the text and added new data to strengthen the conclusions. Therefore, I am satisfied with the response to my comments.

Reviewer #3 (Remarks to the Author):

The authors have adequately addressed all the points I raised in my initial review.

Reviewer #1 (Remarks to the Author):

The revised manuscript addressed most of the criticisms raised in the first review. I recommend this manuscript to be published with a few minor improvements:

1. It is unclear why quantification of miRNA depletion is particularly difficult. Readers should be able to evaluate miRNA level changes in XPO5 KO if the processed data to graph Fig. 8d is given. *We stated the difficulty to quantify miRNA depletion by small RNA sequencing. It is due to the normalization issues and variable ligation efficiency inherent to small RNA sequencing. Nevertheless, we now generate an excel table (Supplementary Data 3) to provide all reads number for miRNAs and other small RNAs detected. In addition, we have already used qPCR and Northern to document the depletion of miRNAs in the manuscript. We are confident readers can extract any information they may needed.*

2. In Fig.5j-related Main text or Methods section, the authors should define “repeat associated regions/RNA”.

The “repeat associated regions/RNA” annotations are peaks that intersect with repeatMasker annotations from UCSC genome browser. We now add this into the method section.

Reviewer #2 (Remarks to the Author):

The authors have addressed all points that I had raised on the previous version of the manuscript. They have clarified parts of the text and added new data to strengthen the conclusions. Therefore, I am satisfied with the response to my comments.

We thank the reviewer for his/her positive response.

Reviewer #3 (Remarks to the Author):

The authors have adequately addressed all the points I raised in my initial review.

We thank the reviewer for his/her positive response.